# Distinct patterns of connectivity with the motor cortex reflect different components of sensorimotor learning

**Corson N. Areshenkoff**[1,2]*, **Anouk J. de Brouwer**[1], **Daniel J. Gale**[1], **Joseph Y. Nashed**[1], **Jonathan Smallwood**[2], **J. Randall Flanagan**[1,2], **Jason P. Gallivan**[1,2,3]

**1** Centre for Neuroscience Studies, Queens University, Kingston Ontario, Canada, **2** Department of Psychology, Queens University, Kingston Ontario, Canada, **3** Department of Biomedical and Molecular Sciences, Queens University, Kingston Ontario, Canada

* c.areshenkoff@queensu.ca

**Data Availability Statement:** Behavioral and imaging data (including T1w and functional scans) are posted to the following repository at

## Abstract

Sensorimotor learning is supported by multiple competing processes that operate concurrently, making it a challenge to elucidate their neural underpinnings. Here, using human functional MRI, we identify 3 distinct axes of connectivity between the motor cortex and other brain regions during sensorimotor adaptation. These 3 axes uniquely correspond to subjects' degree of implicit learning, performance errors and explicit strategy use, and involve different brain networks situated at increasing levels of the cortical hierarchy. We test the generalizability of these neural axes to a separate form of motor learning known to rely mainly on explicit processes and show that it is only the Explicit neural axis, composed of higher-order areas in transmodal cortex, that predicts learning in this task. Together, our study uncovers multiple distinct patterns of functional connectivity with motor cortex during sensorimotor adaptation, the component processes that these patterns support, and how they generalize to other forms of motor learning.

## Introduction

Many of our daily actions stem from the interplay of different mental processes that operate concurrently to achieve our intended goals. Broadly, these mental processes can be distinguished by the nature of their cognitive demands: Explicit processes involve conscious mental operations, often requiring our directed attention and deliberate thought; by contrast, implicit processes operate autonomously, operating beneath the threshold of our conscious awareness [1]. Extensively examined within the realms of psychology and neuroscience, these processes often operate in tandem to accomplish a wide range of tasks, from learning a foreign language to playing a musical instrument or driving a car [2–4].

In recent years, there has been significant interest in understanding how these 2 separate processes support motor learning. For decades, motor learning was widely believed to constitute a singular, implicit process of the sensorimotor system [5,6]. However, recent behavioral and computational work has demonstrated that motor learning is actually supported by at

OpenNeuro (https://openneuro.org/datasets/ds005598) Code availability Imaging data were preprocessed using fmriPrep 1.4.0, which is open source and freely available. Operations on covariance matrices, including estimation and centering, were performed using the R package spdm, which is freely avail- able in a repository at https://zenodo.org/records/13959019. Tutorial code and data for implementing the centering procedure and gra- dient analyses are hosted in a GitHub repository at https://github.com/areshenk-opendata/2023-implicitexplicit. Archived code for performing the analyses can be found at https://zenodo.org/records/14029185.

**Funding:** This work was supported by a Natural Sciences and Engineering Research Council (NSERC) graduate award to C.N.A, and by a NSERC Discovery Grant (https://www.nserc-crsng.gc.ca/index_eng.asp; grant number: RGPIN-2017-04684) and Canadian Institutes of Health Research Grant (https://cihr-irsc.gc.ca/e/193.html; grant number: PJT175012) awarded to J.P.G. The funders had no role in study design, data collection and analysis, decision to publish, or preparation of the manuscript.

**Competing interests:** J.P.G. and D.J.G. are employees of Voxel AI Inc. The other authors report no conflicts of interest. These funding sources had no role in the design, management, data analysis, presentation, or interpretation and write-up of the data.

**Abbreviations:** CSF, cerebrospinal fluid; CW, clockwise; DAN, dorsal attention network; DLPFC, dorsolateral prefrontal cortex; DMN, default mode network; EPI, echo-planar imaging; FD, framewise displacement; FWHM, full-width half-maximum; GM, gray-matter; ICA, independent component analysis; INU, intensity non-uniformity; iPAT, integrated parallel acquisition technologies; IPL, inferior parietal lobe; PCC, posterior cingulate cortex; ROI, region of interest; RT, response time; VMR, visuomotor rotation; WM, white-matter.

least 2 separate learning systems that operate in parallel: a slow-acting implicit system that adapts gradually, and a fast-acting explicit system that adapts rapidly [7–13], and that may depend on brain areas located outside the sensorimotor system. Notably, as performance during learning can reflect the summed contribution of both processes, the ability to characterize their underlying neural systems has remained a challenge. This is because these processes not only need to be disentangled from each other, but also because they must be dissociated from neural activity related to other facets of performance (e.g., sensory feedback related to errors).

Previous research has often framed implicit sensorimotor learning within optimal feedback control theory, with general agreement that such learning reflects the updating of an internal model that links motor commands to sensory outcomes. This form of learning is thought to be supported by a network of cerebellar and sensorimotor cortical regions, crucial for the sensory-guided control of movement [14–16]. In contrast, the neural foundations of explicit learning are considerably less well understood, and may well depend upon regions of cortex that serve more general functions related to cognition, such as association cortex [17]. For example, explicit processes such as strategy use have been associated with the activity of prefrontal cortex [18]; however, it is unclear whether this activity reflects explicit learning per se or the attendant outcomes of using those processes (e.g., the resulting change in performance errors). This is an important neural distinction for understanding how we guide complex behavior, since explicit processes are often brought to bear at critical moments during learning, and can result in a rapid reduction in visuomotor errors [13].

Here, using functional MRI to study whole-brain patterns of functional connectivity with motor cortex, we sought to disentangle these different processes associated with sensorimotor learning. We used subjects' learning behavior in combination with their reports of explicit strategy use to identify the relative contributions of explicit and implicit processes to learning, as well as the trajectory of errors that they make during the task. These behavioral measures were used to identify 3 distinct axes of whole-brain connectivity with the motor cortex that emerged during early learning and that corresponded to subjects' implicit learning, their performance on the task (error rate), and their use of explicit strategic processes, respectively. We found that these separable neural axes were approximately situated at different hierarchical levels of cortical organization; implicit learning was associated with the connectivity of areas in superior parietal and premotor cortex, whereas subjects' visuomotor performance was associated with the connectivity of a subset of regions in the frontoparietal control network. Notably, we found that explicit learning was associated with the connectivity of a subset of areas positioned at the apex of the cortical processing hierarchy, in the default mode network. We tested the generalizability of these separate neural axes by studying sensorimotor learning in a different task—performed in the same subjects—in which abrupt changes in performance are known to rely mainly on explicit strategic processes [19,20]. Together, our findings dissociate 3 distinct patterns of connectivity with motor cortex that correspond to separate neural processes that operate concurrently during sensorimotor learning.

## Results

### Learning behavior during sensorimotor adaptation

To study the neural processes that support sensorimotor learning, we had subjects ($N = 36$) perform a visuomotor rotation (VMR) task [21] in the MRI scanner. In this classic task, subjects were required to move a cursor, representing their right finger position, to intercept a target that could appear in one of 8 locations on a circular ring, using an MRI-compatible touchpad (Fig 1. After a baseline block (40 trials), subjects performed a learning block (160 trials) in which the correspondence between finger motion on the touchpad and movement of

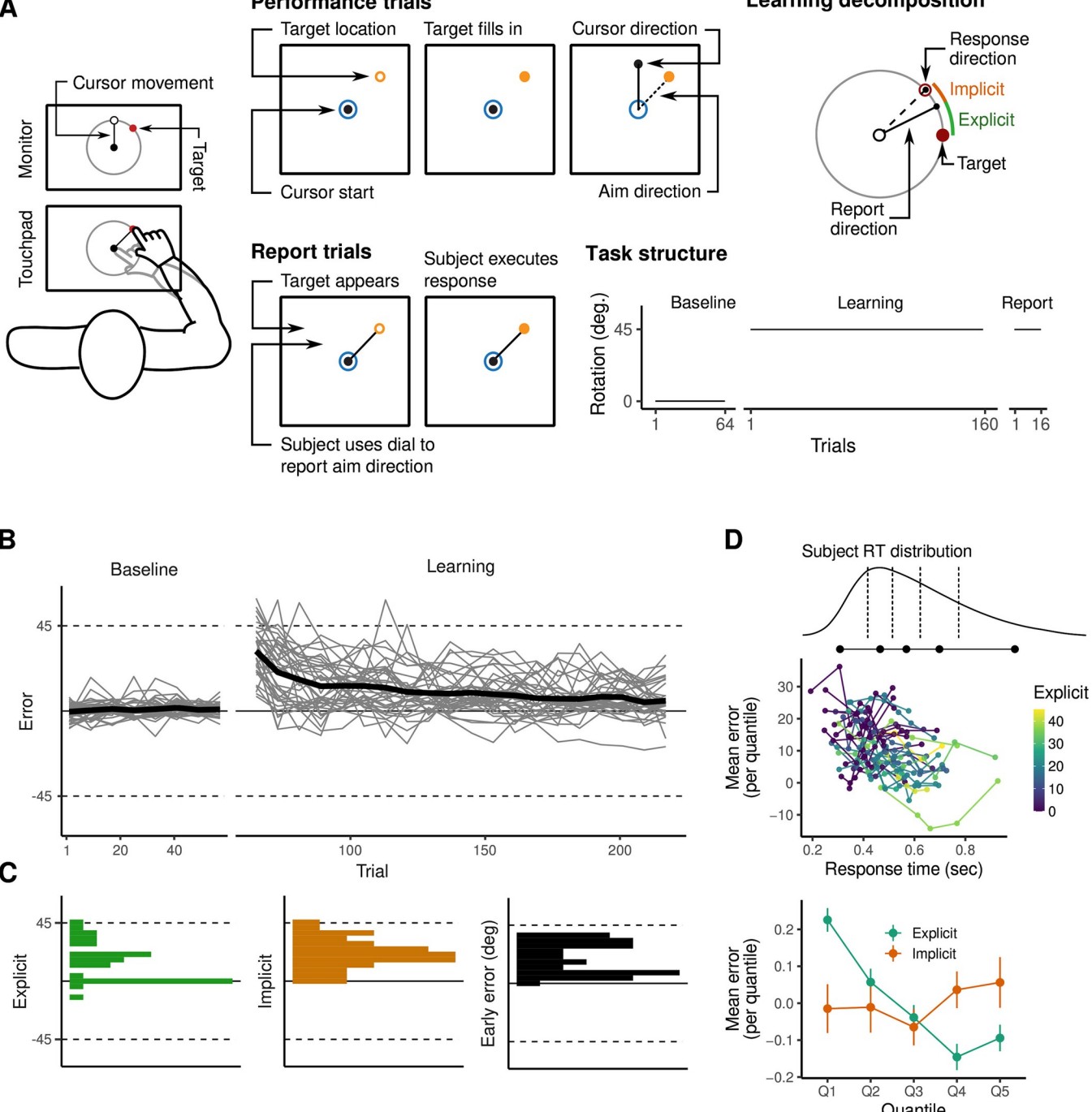

**Fig 1. VMR task and subject behavior.** (**A**) Subjects were cued to move a cursor to a target appearing on a circular ring using an MRI compatible touchpad. During learning trials, the cursor moved at a 45° angle relative to the hand, so that subjects had to learn to adjust their movement direction in order to contact the target successfully. After learning, subjects performed a set of "report" trials in which they used a dial to indicate their aim direction prior to executing the movement. Using these report trials, we derived estimates of subjects' total explicit and implicit learning during the task. (**B**) Error curves for individual subjects. Black trace in bold denotes overall mean error across subjects. (**C**) Distributions of subjects' estimated explicit (top) and implicit (middle) contributions to performance, as well as early error (bottom; defined as the mean error during the first 32 trials). (**D**) For each subject, we computed the mean error during the learning block in each of 5 response time bins separated by the 20%, 40%, 60%, and 80% percentiles. Consistent with previous literature suggesting the use of mental rotation strategies by explicit learners [23], subjects who reported a nonzero re-aiming direction showed monotonically decreasing error with longer response times. The data and code needed to generate this figure can be found in https://zenodo.org/records/14029185.

the cursor was rotated clockwise by 45°. Thus, subjects needed to learn to adapt their movements by applying a 45° counter-clockwise rotation in order to hit the target. At the conclusion of each trial, subjects received visual (error) feedback consisting of both the target location, as well as the position of their cursor, on the target ring. Following the learning block, subjects performed 16 "report" trials [13], in which they were asked to indicate their aim direction using a dial, controlled by their left hand, prior to executing a target-directed movement via their right hand. These report trials were presented at the end of the task, after learning had taken place, in order to avoid drawing attention to the nature of the visuomotor perturbation and thus biasing subjects' learning behavior [22]. Critically, these report trials provided us with a direct index of subjects' explicit knowledge of, and strategy in counteracting, the visuomotor rotation [13].

Behavioral data associated with the VMR task are shown in Fig 1. We found that subjects, on average, were able to successfully learn the VMR task by aiming their hand in a direction that counteracted the visuomotor rotation (Fig 1B). Consistent with prior work [13], we defined subjects' explicit learning as the mean difference between subjects' reported aim direction and the target location, and defined implicit learning as the mean difference between subjects' reported and actual aim directions. Finally, we defined task performance as subjects' average error during the early learning period (defined as the first 100 imaging volumes, or approximately 32 trials, of the task; this was chosen for consistency with our previous work in [24], but see S5 Fig for a robustness analysis using different window sizes). The distributions of these 3 scores are depicted in panel C. Note that we focused our analysis on this early learning period due to evidence that explicit processes are established during this early learning period. Consistent with previous literature [25], we observed a correlation between subjects' reported explicit knowledge and mean response times (RT) during both early ($r = 0.45$, $t = 2.4$, $p = 0.006$) and late learning ($r = 0.69$, $t_{34} = 5.63$, $p = 2.57e − 06$); typically believed to reflect the additional time taken to implement a re-aiming strategy [22,25–27]. As an additional validation of our explicit reports, we further examined the relationship between subjects' error and their RTs. Explicit strategies in the VMR task often make use of mental rotation, and so lower errors are typically observed with longer response times [23]. For each subject, we computed the mean error during the learning block in each of 5 bins, defined using the quantiles of their response time distribution (Fig 1D). Using these bins, we computed the (Spearman) correlation between error and RT for each subject, and observed a significantly lower (negative) average correlation within explicit (compared to implicit) learners ($m_{exp} = −0.44$, $m_{imp} = 0.11$, $t_{3}4 = −2.38$, $p = 0.033$). That is, explicit learners were more accurate at longer response times, while the accuracy of implicit learners did not vary with response time.

Our decision to elicit subjects' reports at the end of the learning block was based on evidence that report trials may make subjects aware of the visuomotor rotation and predispose them to use an explicit strategy during the task [22,28]. Nevertheless, group-level behavioral studies suggest that subjects' explicit re-aiming strategies may evolve during learning the process (e.g., have a larger magnitude during early than late learning; [10,13]). As such, reports collected at the end of the task may not directly reflect the re-aiming strategy subjects used during the early learning process itself. To address this possible concern, we conducted a separate behavioral experiment outside of the MRI scanner ($N = 14$), in which report trials were interspersed continuously throughout the learning block (see S1 Fig). Importantly, we found that subjects' explicit reports collected in the final trials of the learning block were highly correlated with reports collected during the early learning block ($r = 0.77$, $t_{12} = 4.22$, $p = 0.0012$; S1C Fig). Further, we found that subjects' typically developed explicit awareness of the rotation early in the learning block, and that their subsequent explicit reports were highly stable throughout the VMR task (S1D Fig). Taken together, these findings suggest that the late

reports collected during our MRI testing session serve as a valid proxy for the explicit strategy developed during this early learning period.

## Connectivity-based approach for studying sensorimotor adaptation

To examine the neural bases that support explicit and implicit learning, as well as the encoding of performance during the task, we examined learning-dependent changes in functional connectivity between the primary motor cortex and the rest of cortex, striatum, and cerebellum (Fig 2A). By constraining our analysis to bipartite functional interactions with primary motor cortex, we were specifically interested in assessing how various regions in the cortex, striatum, and cerebellum, modulate their connectivity with cortical areas positioned in the final motor output pathway that produce motor commands resulting in the measured behavioral adjustments. For each brain region, we extracted fMRI timecourse activity from different atlas

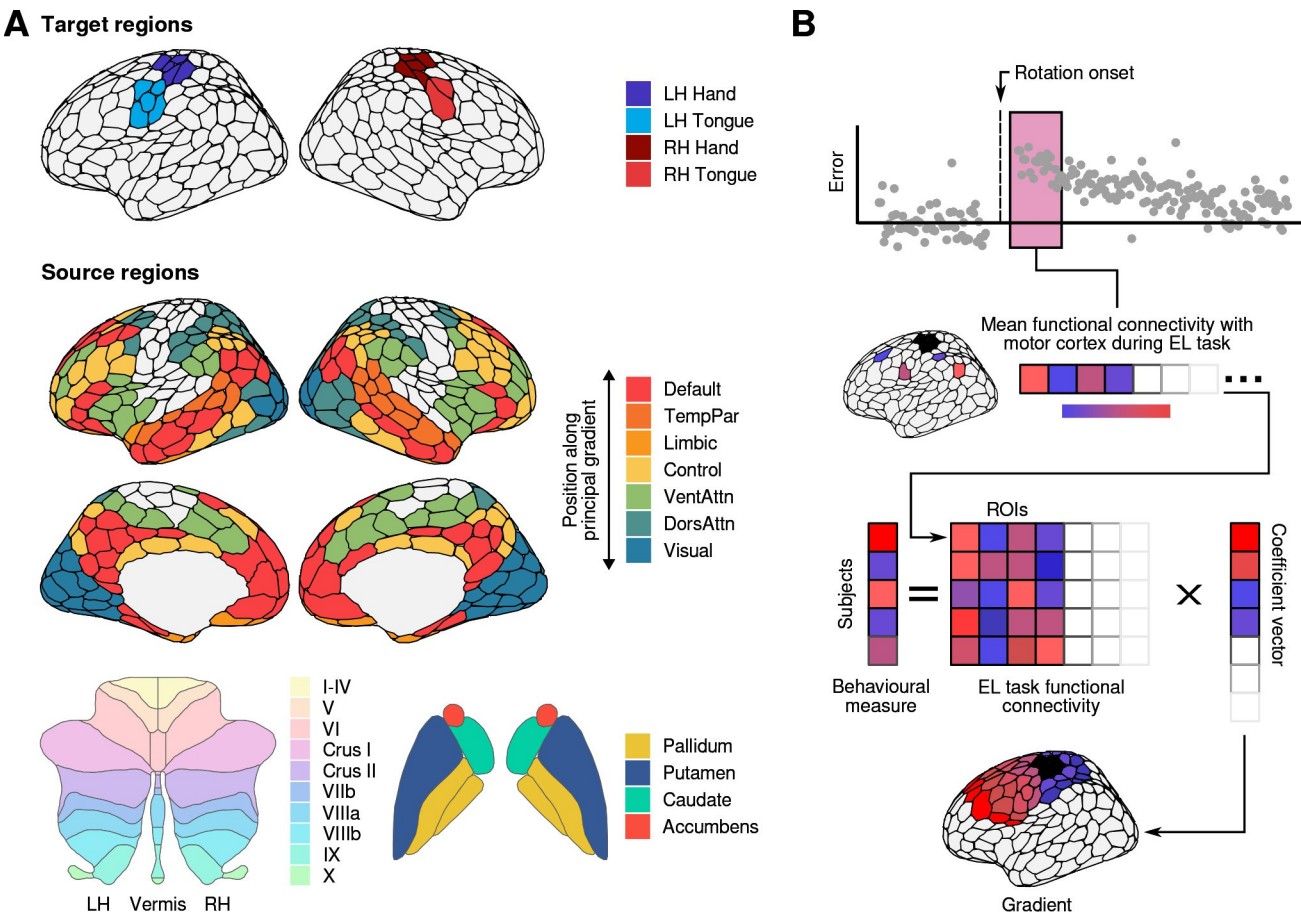

**Fig 2. Brain networks/regions used in our neural analyses and predictive model.** (**A**) We studied functional connectivity between primary motor cortex and the rest of cortex, striatum, and cerebellum during VMR learning. Cortical networks were derived using the 400 region parcellation of [29] and assigned to the 17 networks classified by [31]. Here, for the cortical regions, we have arranged the network names according to their position along the principal gradient of functional cortical organization demonstrated by [32]; with networks more proximal to primary sensorimotor regions at the lower-end of the cortical gradient (bottom), and networks more distant from sensorimotor regions at the higher-end of the gradient (top). Cerebellar regions were identified using the atlas of [33]), while striatal regions were extracted from the Harvard-Oxford atlas. Motor cortex parcels were identified as belonging to either hand or tongue areas of the left and right hemispheres based on the reported findings of [29]. (**B**) We used a set of penalized regression models (see Methods) to predict subjects' explicit learning, task performance, and implicit learning from patterns of whole-brain functional connectivity with primary motor cortex. The coefficients from each of the fitted models produce a set of brain maps which we hereby refer to as the explicit, performance, and implicit neural axes. The data and code needed to generate this figure can be found in https://zenodo.org/records/14029185.

parcellations of the cerebellum, striatum, and all of cortex (see Methods), and studied functional connectivity between these brain regions with the average activity of a subset of 4 regions in the left (contralateral) hemisphere comprising the hand motor region (belonging to the left Somatomotor A network; [29]). For each subject, we estimated covariance networks during the baseline, early learning, and late learning periods (each defined as an epoch of 100 imaging volumes), allowing us to describe changes in connectivity over the course of the task. Given that a large body of recent literature suggests that functional connectivity is dominated by static, subject-level differences that can obscure any task-related variance [30], we first centered subjects' covariance matrices to align their mean covariance across resting-state, baseline, and learning ([24], see Methods for details and for a more thorough description). These centered covariance matrices were then used to identify patterns of functional connectivity associated with distinct learning processes.

Note that, because both implicit and explicit learning result in better performance (less error), the neural correlates of these measures may also reflect effects due to the experience of errors. For example, systems involved in the goal-directed deployment of spatial attention may be necessary for successful task performance regardless of the specific mechanisms by which a given subject learns, and so may be recruited by both implicit and explicit learners. Thus, in order to derive relatively "pure" measures of these 2 learning processes, we controlled for subjects' performance by using the mean error during the early learning epoch as a covariate in our model. Specifically, we separately regressed each of our explicit and implicit learning measures, and the functional connectivity of each ROI with motor cortex, onto subjects' early errors, and used each of the residuals for further analysis. Separately, in order to derive neural activity patterns linked to learning performance per se, rather than any particular learning process, we also derived patterns of functional connectivity associated with subjects' visuomotor errors. We interpret this "Performance" axis as reflecting processes related to general error-detection, or to general task-based processes which may be common to both forms of learning.

Using the patterns of functional connectivity across regions extracted from the early learning period, we used a group-penalized ridge regression model to predict subjects' explicit and implicit learning, as well as their performance errors ([34]; see Fig 2, right, and see Methods). We initially focused our analysis on this early learning period as this is the point in time in which explicit learning develops ([13], see also S2 Fig). We used the region-wise coefficients from this model to create whole-brain maps reflecting covariance with motor cortex during early learning, which we hereby refer to as the Explicit, Implicit, and Performance neural axes (Fig 3). Note that we inverted the sign of subjects' error in the derivation of the performance axis, so that greater loadings were associated with greater performance (i.e., lower error). For each axis, network-specific penalties were tuned by leave-one-out cross-validation (cross-validated $R^2$ : $R^2_{exp} = 0.29, R^2_{perf} = 0.82, R^2_{imp'} = 0.64$).

## Separate neural axes are associated with different component processes of sensorimotor adaptation

Within the Implicit neural axis, we observed the highest regional loadings in the dorsal attention network (DAN), and specifically the DAN-B subnetwork, comprising superior parietal and premotor regions known to support the sensory-guided control of movement [14–16]. Note that, although we did not additionally observe high regional loadings in the cerebellum, a separate analysis focused solely on the cerebellum returned results consistent with its role in implicit error-based adaptation; see Discussion and S3 Fig. By contrast, within the Performance neural axis, we observed the highest regional loadings within the Control network, and specifically the Control-A subnetwork, mainly comprising the dorsolateral prefrontal cortex

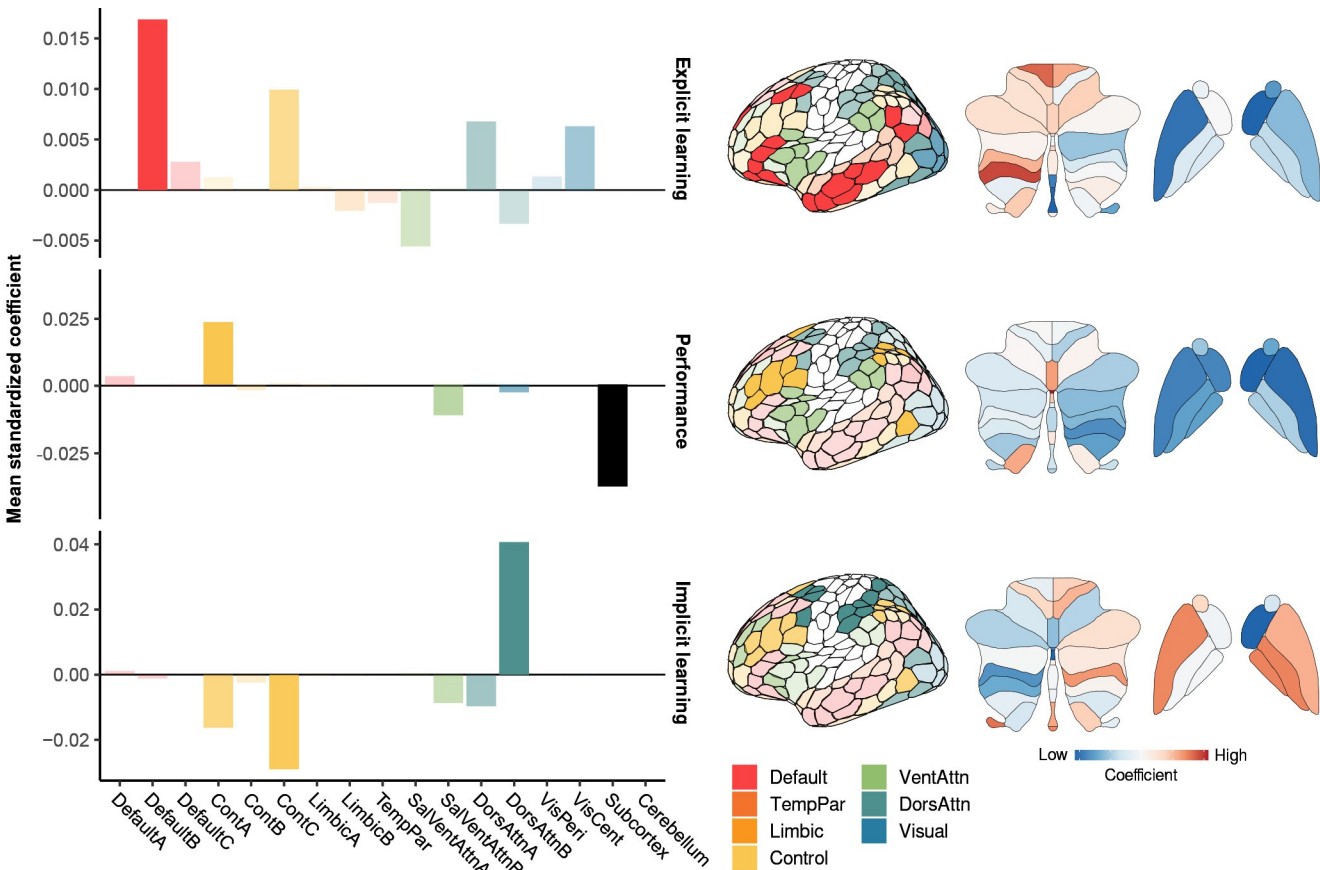

**Fig 3. Explicit, performance, and implicit neural axes.** (Left) Barplots show mean coefficients from our penalized ridge regression model for each subnetwork [30]. (Right) The most prominent subnetworks for each neural axis are displayed on cortical surfaces. Loadings for cerebellar and striatal regions are shown at the right; note that colormaps for each surface are scaled individually. The data and code needed to generate this figure can be found in https://zenodo.org/records/14029185.

(DLPFC), inferior parietal lobe (IPL), and anterior cingulate cortex (see barplots in Fig 3). In the sensorimotor adaptation literature, the lateral prefrontal cortex has been shown to increase its activity during early learning [18,35,36], and thus it has been suggested to play a key role in developing explicit strategies during learning [37,38]. Critically, however, for the Explicit neural axis we actually observed the highest loadings within the default mode network (DMN); specifically, within the DMN-B subnetwork, comprising higher-order regions in the middle and inferior frontal gyrus, angular gyrus, and anterior and middle temporal cortex (see barplots in Fig 3).

Note that, due to the sparsity of the coefficients returned by our models, we found that most cortical networks loaded only on a single axis (e.g., the DAN and DMN networks loaded almost exclusively on the Implicit and Explicit neural axes, respectively; Fig 4A). However, as a noteworthy departure from this general observation, we found that the frontoparietal control network exhibited substantial cross-loadings on the different neural axes. For example, Fig 4A (bottom right) shows the implicit/explicit loadings of individual regions within the Control subnetworks. As can be seen in this plot, it was solely the Control-A and Control-C networks that showed meaningful cross-loadings across both axes, with the posterior cingulate cortex (PCC) and precuneus regions, in particular, being strongly explicit-aligned. In this way, a

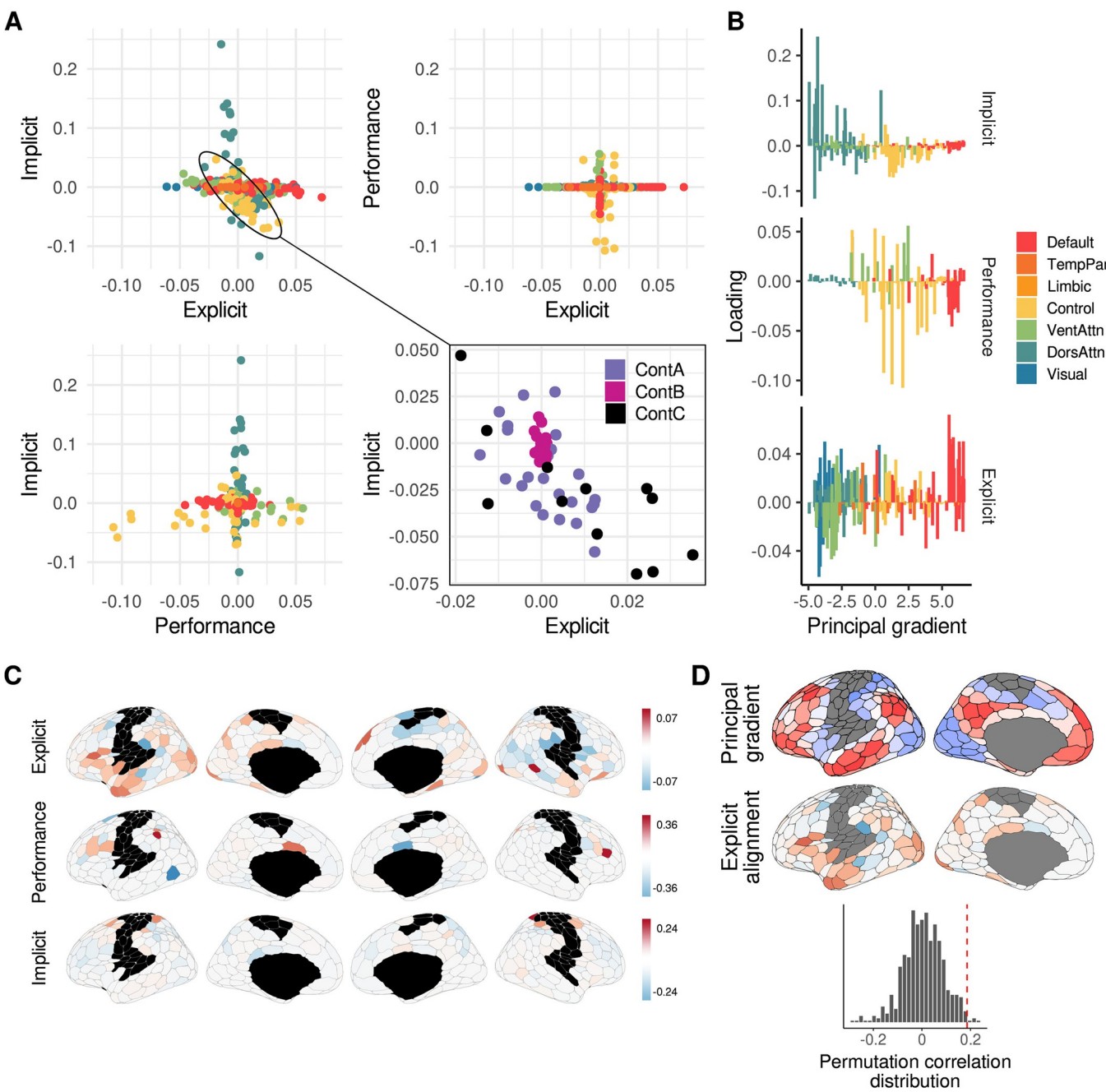

**Fig 4. Alignment of individual cortical regions along the 3 separate neural axes.** (**A**) Relationship between the loadings of individual cortical regions for the Explicit, Performance, and Implicit neural axes. Note that only Control regions showed meaningful cross-loading between axes (highlighted by ellipse). These regions are shown separately in the bottom right panel. Control C subregions (posterior cingulate and precuneus) in particular were strongly aligned to the explicit axis. (**B**) Loadings of individual brain regions for each neural axis, plotted according to their positioning along the Principal gradient. (**C**) Surface maps of the 3 neural axes. Red colors denote a positive loading and blue colors denote a negative loading. (**D**) Relationship between the Principal gradient and Explicit-alignment axis. Plot at bottom shows the null distribution of spatial correlations between the Principal gradient and Explicit-alignment axis, as derived from 1,000 iterations of the spatial autocorrelation-preserving null model [39] implemented by the Neuromaps toolbox [40]. The dashed red vertical line denotes the true correlation value ($p = 0.028$). The data and code needed to generate this figure can be found in https://zenodo.org/records/14029185.

particular brain region can be described as being either explicit- or implicit-aligned if it has a greater loading on the Explicit or Implicit neural axis, respectively (we return to this idea in the next section).

## The learning-related neural axes map onto different levels of functional cortical organization

In order to better understand the large-scale organization of the different neural axes, we took the loadings of individual regions on each axis (Fig 4, panel B) and plotted them according to their position along the principal gradient of functional brain organization derived by [31] (Fig 4, panel C, see panel D for a visualization of the principal gradient). This principal gradient, which spans from unimodal cortex (i.e., visual and somatomotor networks) to transmodal cortex (i.e., the DMN), signifies a hierarchy in cortical processing from lower- to higher-order systems [17,31,41]. As can be seen in Fig 4, panel C, regions loading on the implicit axis tend to be situated lower on the principal gradient (i.e., closer to sensory and motor regions), while the performance axis comprises regions closer to transmodal cortex. By contrast, the explicit axis prominently features regions of the default mode network, located neuroanatomically furthest from sensorimotor cortex.

Next, to more directly assess the relationship between our learning-derived axes and this principal gradient of functional cortical organization, we derived a single summary axis—the "explicit alignment" axis—by computing the difference in rank loadings for each ROI (Fig 4D). According to the construction of this summary axis, regions with positive values indicate a relatively greater loading on the explicit, compared to implicit, neural axis, and vice versa for negative values. Consistent with this, the greatest explicit alignment is observed in the DMN-B and Control-C subnetworks, whereas the least explicit (most implicit) alignment is observed in the superior parietal and premotor areas of the DAN-B subnetwork. Notably, we found this explicit alignment axis was significantly correlated with the aforementioned principal gradient of functional brain organization ($p = 0.028$; spin permutation test: [42], Fig 4D). This suggests that explicit learning is supported by brain areas (i.e., transmodal regions) positioned at the very apex of the cortical processing hierarchy, whereas implicit adaptation is supported by brain areas (superior parietal and premotor regions) located directly adjacent to sensorimotor cortex.

## Expression of the explicit neural axis predicts learning in a separate reward-based motor task

Implicit adaptation in the VMR task is thought to primarily reflect the updating, through sensory prediction errors, of an internal forward model that predicts the sensory consequences of a motor command [5,7]). In other forms of motor learning, where subjects are denied this sensory error feedback, this same kind of updating is impossible, and so learning in such cases must rely on alternative neural systems and pathways [43,44]. By contrast, explicit learning has been linked to various executive functions [18,38,45,46], and the DMN—which features prominently in our Explicit neural axis—is suggested to specialize in cognitive computations that are independent of specific perceptual inputs [41]. We thus reasoned that the Explicit neural axis might generalize across motor learning tasks that provide different forms of sensory feedback, whereas the implicit neural axis would not.

To test this idea, in a different fMRI session we had our same participants perform a separate motor learning task in which they were required to alter their hand movements purely through reward-based feedback. In this task, subjects used their right finger on the touchpad to trace—without visual feedback of their cursor—a rightward-curved path displayed on the screen (from a start position to target line). Following a baseline block (70 trials), in which subjects did not receive any feedback about their performance, they performed a learning block (200 trials). Here, they were instructed that they would now receive reward score feedback, between 0 and 100, based on how accurately they traced the visible path, and that they should

attempt to maximize their score across trials. Critically—and unbeknownst to the subjects—the reward score they received was actually based on how accurately they traced the (hidden) mirror-image path (reflected across the vertical axis; i.e., they would receive an imperfect score if they traced the visible path but would receive a 100 score if they perfectly traced the mirror image path; Fig 5A). Importantly, as the cursor was invisible, subjects could not use error-based learning mechanisms to improve their performance (as they could in the sensorimotor adaptation task). Rather, they could use only the scalar reward feedback, presented at the end of each trial, to refine their movements over time.

Learning in such tasks—i.e., where behavior must be guided solely by scalar feedback—is known to be highly explicit [19,20,47–49]. Consistent with this view, we found that the vast majority of subjects displayed clearly defined learning periods during our task, book-ended by periods of relatively flat performance (see example subjects in Fig 5). To identify these learning periods in a data-driven fashion, on each trial we computed a movement score quantifying the overall leftward (towards the invisible target) or rightwardness (towards the trained path) of the trajectory, and fit a sigmoid function to these scores across trials. These resulting fits allowed us to define separate pre-learning, learning, and post-learning periods (see Fig 5C; see Methods and see fits to individual subjects in S4 Fig). We then asked whether the expression of the Explicit neural axis—specifically during the learning period—was related to subjects' rate of learning. To test this, we performed a two-part analytical procedure: (1) For each subject, we computed the spatial correlation between each of our learning-related neural axes extracted from the VMR task (i.e., the Implicit, Performance, and Explicit axes) with the observed patterns of functional connectivity during each of the separate periods (pre-learning, learning, and post-learning) associated with the reward-based learning task. [Note that these spatial correlations provide us with a direct index of the degree to which each neural axis from the sensorimotor adaptation task is "expressed" during the different periods of reward-based learning, and we thus call these correlations the explicit (resp. performance, implicit) axis scores.] (2) We examined whether these newly derived axis scores were correlated with subjects' learning rates. The results of this analysis are shown in Fig 6B.

Consistent with our predictions, we observed a significant positive correlation between the expression of the Explicit neural axis (derived from the VMR task) during the learning period of the reward-based task, with subjects' learning rates. That is, the larger the increase in the expression of the explicit component during the learning phase, the more rapidly subjects learned the hidden path ($r = 0.34$, $t_{34} = 2.04$, $p = 0.049$). Critically, as an important control analysis, we found that this across-task prediction was unique to the Explicit neural axis, as we observed no such relationship for either of the Implicit ($r = −0.12$, $t_{34} = −0.71$, $p = 0.48$) or Performance ($r = −0.05$, $t_{34} = −0.29$, $p = 0.77$) neural axes during the same learning period. Moreover, as a further control, we also observed no correlation between the expression of the Explicit axis and performance in the reward learning task when we separately examined both the pre-learning ($r = −0.06$, $t_{34} = −0.34$, $p = 0.74$) and post-learning periods of the task ($r = −0.18$, $t_{34} = −1.01$, $p = 0.32$). Together, these results strongly suggest that the brain systems comprising our Explicit axis are recruited specifically during the learning period itself—when subjects are actively adjusting their behavior—rather than in the preceding and subsequent periods of relatively stable performance, respectively.

One alternative explanation of the above results is that, rather than the Explicit neural axis uniquely reflecting a modulation of motor regions involved in response generation, they instead reflect a more global, whole-brain change in functional connectivity. To test this alternative explanation of the results, we performed an additional set of control analyses in which we repeated the construction of the 3 axes (Explicit, Implicit, and Performance) using patterns of functional connectivity with the contra- and ipsilateral (to the response hand) tongue area

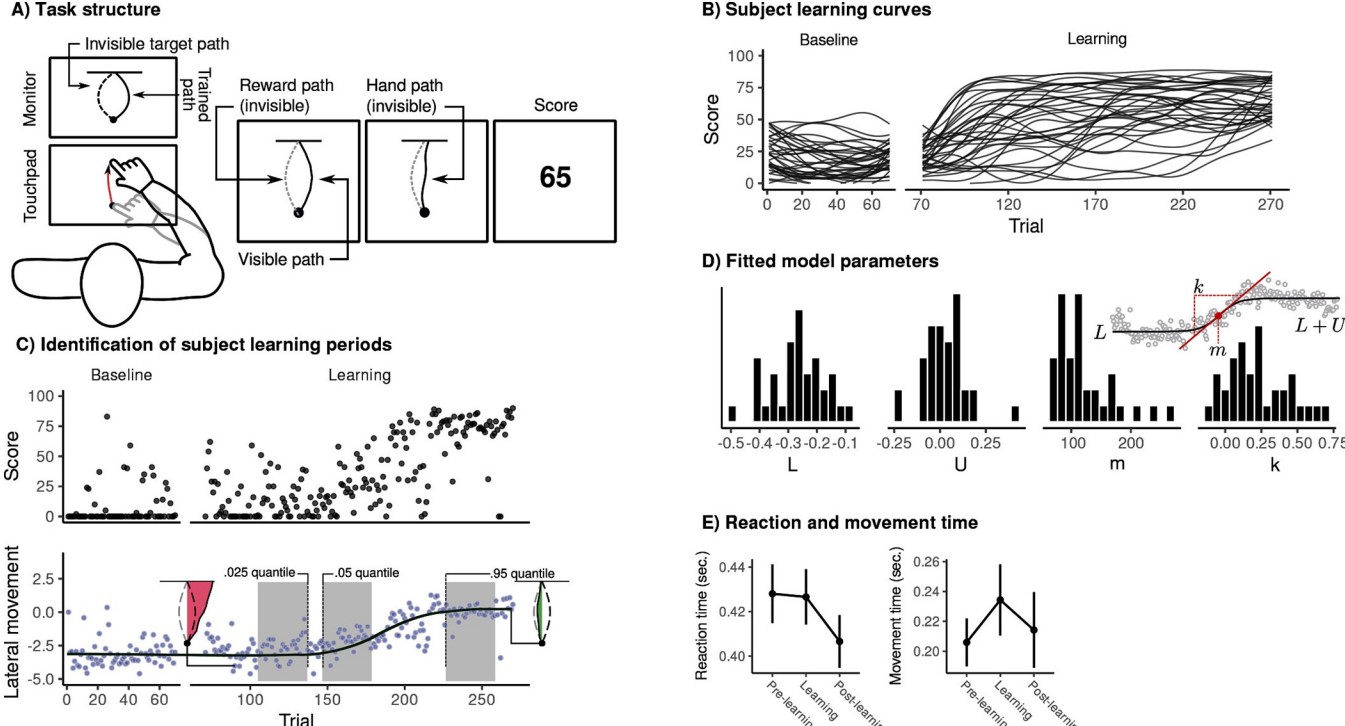

**Fig 5. Reward-based motor task and subject performance.** (**A**) Subjects learned to trace an invisible target trajectory with only score feedback. After learning to trace a rightward curved path (baseline trials), score feedback was introduced indicating (unbeknownst to the subject) their success in tracing the mirror image path (learning trials). (**B**) Spline-smoothed scores for individual subjects. Note the considerable heterogeneity in subjects' learning, with some subjects beginning to learn immediately after score onset (as was the case in the VMR task) versus some subjects showing no improvement until much later. (**C**) Behavioral data from an example subject. (Top) Score across trials. (Bottom) For each trial, we quantified the overall leftward or rightwardness of the response by taking the average x-position of the movement trajectory, with positive values indicating that the response was more towards the invisible target. We described this value as the lateral movement score. Most subjects showed clearly defined learning periods, book-ended by periods of stable performance. We identified these periods by fitting a sigmoid function to each subject's movement scores in order to identify these periods. The 0.025, 0.05, and 0.95 quantiles of the resulting curves were used to define the end of the pre-learning, beginning of the learning, and beginning of the post-learning periods, respectively. In addition, the parameters of the fitted sigmoid were used to characterize subjects' performance. (**D**) Distributions of fitted model parameters across subjects. $L$ denotes the lower asymptote, and $L + U$ denotes the difference between the upper and lower asymptotes, respectively; $m$ denotes the midpoint of the learning period; and $k$ denotes the learning rate. (**E**) Relationship between subjects' reaction time (RT, left) and movement time (MT, right) as a function of learning period. Data shows the group mean +/- 1 standard error of the mean. Note that subjects' RTs and MTs tended to decrease during the post-learning period, consistent with the responses becoming more automatic once subjects have converged upon a stable solution to the task. The data and code needed to generate this figure can be found in https://zenodo.org/records/14029185.

of somatomotor cortex, as well as the ipsilateral hand area during the VMR task. Critically, in none of these cases did we observe any significant correlation between axis expression and performance in the reward-based learning task (Fig 6, panel B). [Note that, although the correlation between Explicit axis expression and learning rate was not significant in the right hand area, neither was the difference in correlation between the 2 hemispheres significant.]

Another alternative explanation of our main results is that Explicit neural axis, instead of reflecting processes related to the learning of an explicit strategy, relates purely to the implementation of the learned strategy (e.g., mental rotation of the response). If this were true, then we would expect to also observe the same pattern of across-task prediction effects when deriving an Explicit neural axis using instead the late learning epoch of the VMR task (the last 4 blocks of 8 trials). Indeed, during this late learning epoch subjects' performance has plateaued and their RTs are decreased (Fig 1D), consistent with an increased automaticity in the motor responses. To test this idea, we again repeated the construction of the 3 neural axes using the patterns of functional connectivity during the late learning epoch of the VMR task, and again

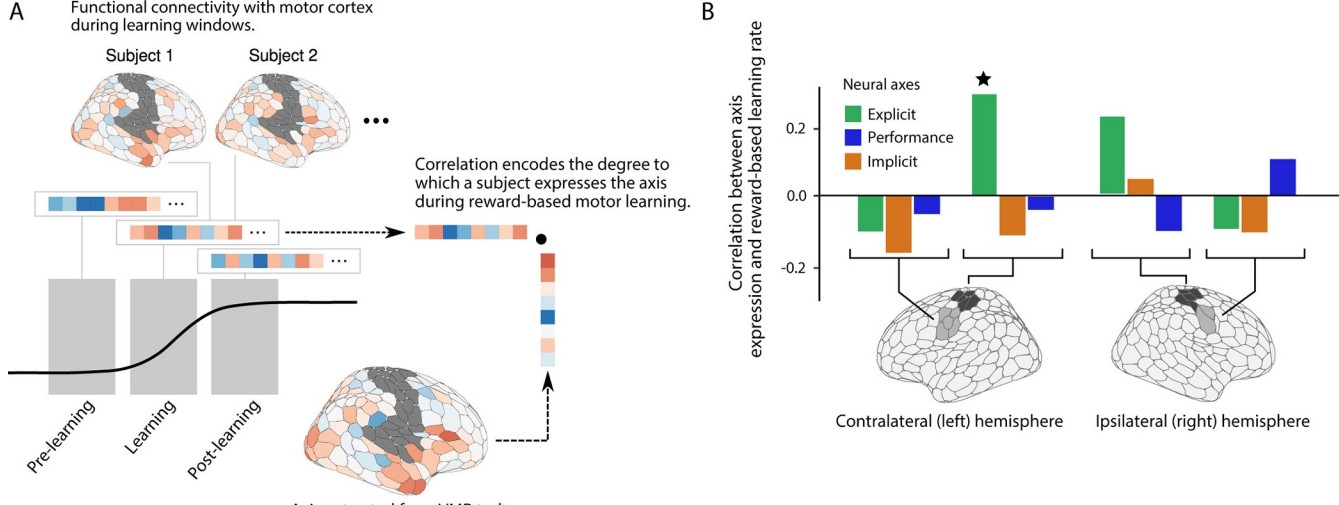

**Fig 6. Neural results for the separate reward-based motor learning task.** (**A**) We examined whole-brain functional connectivity with motor cortex during each period of the reward-based motor task (pre-learning, learning, and post-learning) and asked whether learning-related increases in the expression of the Explicit neural axis was associated with better learning (i.e., $k$ parameters from Fig 5). For each period of the reward-based motor task, we computed functional connectivity with motor cortex and then examined the correlation between these patterns of functional connectivity, and the 3 neural axes derived from the VMR task (shown in Fig 4). These correlations served as measures of the expression of the Explicit, Performance, and Implicit axes during reward-based learning. (**B**) Learning rates, derived from subjects' sigmoid fits, were significantly correlated with the increase in Explicit axis expression during learning, but not with the Implicit or Performance axis expressions. As a control, we repeated our analysis, including the derivation of the VMR axes, using functional connectivity with the ipsilateral (to the response hand) tongue and hand areas of the somatomotor network, as well as with the contralateral tongue area. In each case, the correlation with learning rate was abolished. The data and code needed to generate this figure can be found in https://zenodo.org/records/14029185.

found no correlation between any of these axes' expression and performance in the reward-based learning task (S6A Fig). Note also that the explicit axis derived from subjects' late learning periods was markedly different from its early learning equivalent (S6B and S6C Fig). During late learning, the explicit axis prominently featured regions of the dorsal attention and visual networks—in particular, premotor and posterior parietal regions, as well as central and peripheral visual regions. Although we do not directly assay the specific strategy used by explicit subjects, mental rotation strategies are commonly used by explicit learners in the VMR task [23], and regions of the premotor, posterior parietal, and visual cortex are frequently implicated in mental rotation [50]. This interpretation would be consistent with the failure to observe a correlation between the expression of the late explicit axis and performance in the reward-based learning task, as mental rotation is unlikely to support learning in the reward-based task. Thus, we interpret this late explicit axis as reflecting the implementation of a strategy specific to the VMR task (e.g., mental rotation), while the early explicit axis likely involves components related to the learning of a strategy itself.

Collectively, these control analyses indicate that it is the expression of the Explicit neural axis, and not the Implicit or Performance neural axes, that predicts learning in a separate motor task that does not afford learning via sensory prediction errors. Moreover, these additional analyses demonstrate an important selectivity to this effect—specifically, we find that the across-task prediction is exclusive to (1) subjects' individual learning phase (and not their pre- and post-learning phases); and (2) connectivity with the contralateral hand area of motor cortex, indicating that these patterns of connectivity are both neuroanatomically and behaviorally relevant.

## Discussion

Adaptive motor behavior is achieved through a combination of multiple learning processes, yet the specific functional interactions between brain regions that support these processes are poorly understood. Here, we examined learning-related changes in human cortical, striatal, and cerebellar functional connectivity with the primary motor cortex while individuals performed a classic sensorimotor adaptation task. Using subjects' explicit reports, as well as their visuomotor errors experienced during adaptation, we isolated 3 distinct axes of connectivity during early learning that corresponded to subjects' implicit and explicit learning processes, as well as their visuomotor performance. Whereas we found that implicit learning was related to connectivity changes between motor cortex and superior parietal and premotor regions (the Implicit neural axis), we found that explicit learning was related to connectivity changes between motor cortex and higher-order transmodal cortex, in particular the DMN (the Explicit neural axis). Notably, we distinguished both of these connectivity changes from activity patterns related to subjects' performance errors during learning, which we found was instead related to connectivity changes in the frontoparietal control network (the Performance neural axis). Further analyses showed that these 3 distinct neural axes mapped onto different levels of the cortical processing hierarchy, extending from primary sensory and motor systems to higher-order transmodal cortex. Finally, we examined the extent to which each of these neural axes generalized to a separate motor task known to rely primarily on explicit learning processes and found that it was the expression of the Explicit neural axis—and not the Implicit and Performance axes—that predicted subject learning under these circumstances.

The main contribution of our study lies in establishing a tripartite distinction in how connectivity between the motor cortex and other brain regions relates to implicit learning, explicit learning, and visuomotor performance. Whereas in behavioral studies of motor learning, these different factors can be readily distinguished and measured [7,13,22], disentangling their neural bases has remained a challenge. Indeed, while activation changes observed during early learning may reflect subjects' formation of cognitive strategies—as is often interpreted [24]—these changes may also reflect modulations in sensory feedback associated with the use of such strategies (e.g., decreased errors) or trial-by-trial recalibration of the internal model ([7,13,51,52], which occurs automatically in parallel with explicit learning). Our study addressed these challenges by separately measuring subjects' actual re-aiming strategies and controlling for performance-related effects on explicit and implicit learning measures. This approach enabled us to identify unique neural signatures of these distinct processes through patterns of functional connectivity between the motor cortex and other brain regions.

A key finding from our study is to underscore the important role of the superior parietal and premotor areas within the DAN in implicit adaptation (see Fig 7, bottom). This learning process is frequently understood through the lens of optimal feedback control [53], where the observed response results from a policy dictating the motor control signal generated by primary motor structures. Successful execution of this policy requires an internal model to generate accurate predictions about the sensory consequences of a motor command. Recent perspectives suggest that this model is updated based on perceptual errors, reflecting a mismatch between the target and the perceived movement position, which is derived from a multisensory estimate combined with a motor efference copy [54]. Previous work has implicated regions of the parietal and premotor cortex, as well as the anterior cerebellum, in these processes [55–58]. Note that although our findings identified these same cortical regions (in the DAN), we found that the cerebellum loaded relatively weakly onto our Implicit neural axis. However, we should note that its anterior motor regions—particularly in the right (ipsilateral)

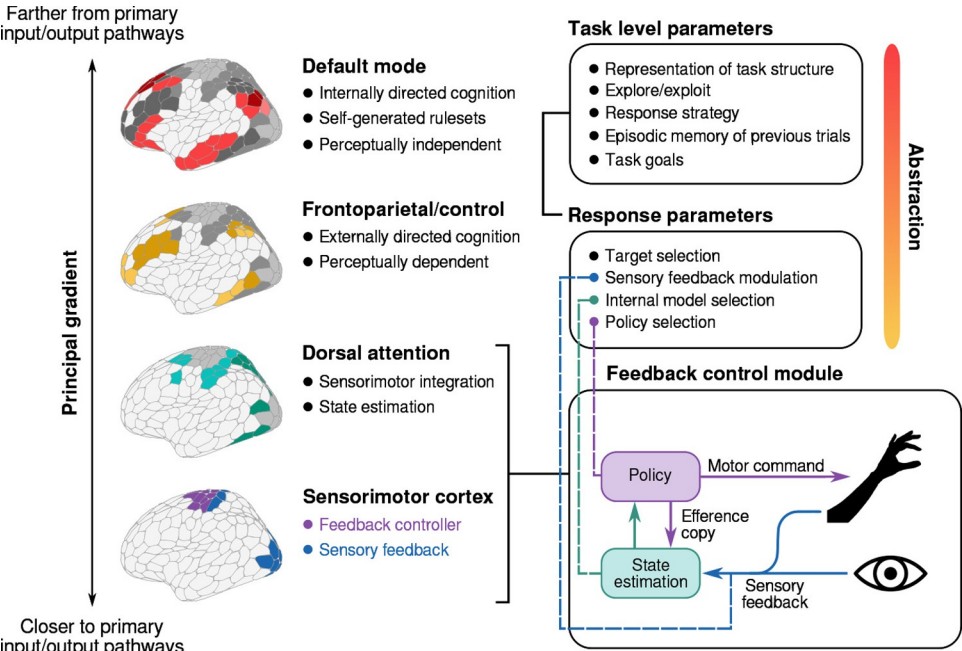

**Fig 7. Conceptualizing the current results according to current neuroanatomical perspectives.** Sensorimotor adaptation is frequently interpreted through the lens of optimal feedback control, in which adaptation results from the updating of an internal forward-model in response to sensory prediction errors [21]. The cortical circuitry supporting such adaptation comprises a network of premotor and parietal regions (along with the cerebellum) [53] commonly grouped within the dorsal attention network, which featured prominently in our Implicit axis. The parameters which dictate the operation of this system (e.g., which target is selected for action) are presumably set by prefrontal regions that are located more distally from primary sensorimotor cortex. At the apex of the cortical processing hierarchy, the default mode network is believed to support temporally extended (e.g., spanning multiple trials), perceptually independent processes which allow responses to be guided by internally generated rule-sets, and by multiple sources of information integrated over longer time periods [41]. It is this latter network which our analysis associated with explicit learning processes.

cerebellum—were aligned to our Implicit neural axis, consistent with its known role in computing the sensory prediction errors that drive implicit motor adaptation [44,59–62].

A second key feature of our results is the distinction between 2 separate patterns of functional connectivity during sensorimotor learning: one linked to explicit learning and the other linked to visuomotor performance. In prior work, explicit learning has been suggested to involve higher-order control regions in the prefrontal cortex, as well as processes like working memory and attention [24,37]. Yet, the specific influence of these control signals on motor system parameters has been unclear. For example, it remains to be formalized whether these control signals dictate the selection of a new policy (i.e., a decision to aim somewhere other than directly at the target), the selection of an alternative forward model, or the weighting of specific sensory inputs based on task demands. Indeed, regions of the prefrontal cortex have been implicated in exactly these types of processes (e.g., [63]) and associated with performance during visuomotor adaptation [10,12,24,37]. However, the extent to which these Control networks specifically contribute to the learning of an explicit strategy per se—versus processes which may support task performance more generally (e.g., goal-directed spatial attention), and thus lead to better performance—has not been systematically studied in previous work. Indeed, Control network areas like the DLPFC have been implicated in processes essential for basic task performance—such as the selection of spatial targets for action and with performance monitoring [64–66]—in addition to any specific role in explicit learning. Our current results

suggest that functional connectivity with the DLPFC (Control-A subnetwork) is not associated with explicit learning specifically, but rather with processes that assist in task performance (e.g., spatial attention, error detection). The fact that expression of this axis was not correlated with learning rate in the separate reward-based motor task may thus reflect the fact that this axis includes (in part) patterns of connectivity associated with the unique demands of the VMR task (e.g., spatial target processes), which are not relevant to performance in the reward-based task, and/or with effects related to the monitoring of performance (e.g., detection of directional errors), rather than processes contributing to the formation of a strategy.

Although our finding that explicit learning was associated principally with regions of the default mode network (DMN) may be unexpected in the context of much of the sensorimotor learning literature, there are numerous findings across diverse research fields hinting at just such a role. For example, recent theories of DMN functioning (e.g., [41]) have implicated the DMN in states such as self-referential processing, theory of mind, and spontaneous states such as mind wandering—all of which depend on explicit conscious attention [17,41,67–70]. In addition, there is emerging neurophysiological evidence that DMN areas are uniquely engaged in conditions when performance is initially poor and when executive control is advantageous [71,72]. For example, work in nonhuman primates has shown that activity in DMN areas is linked to the formation of strategies during periods of learning when errors are high [73]. Further, early neuroimaging work in explicit motor sequence learning has implicated regions of the DMN in task performance, though notably much of this work had preceded the characterization of the DMN as a distributed functional network (e.g., [74]).

In this way, our results suggest that—like in multiple other domains—the DMN may play a role in motor control that is linked to the capacity of humans to explicitly attend to their own thoughts and actions. This skill is particularly important when behavior must be guided by internally generated strategies and rule-sets, and by information accrued over longer time horizons than by in-the-moment sensory inputs [41]. This contribution of the DMN to behavior is hypothesized to be enabled through its unique topographic positioning on cortex [31], with each core node of the DMN being located maximally distant from sensory and motor systems. This topography is thought to free DMN areas from the constraints of extrinsically driven neural activity, thus allowing these regions to play a role in the orchestration of information processing across distributed functional systems [41,75].

Indeed, as noted in our results, we found that our 3 neural axes are situated at increasing positions along the cortical processing hierarchy that spans from unimodal to transmodal cortex (see Fig 7). The implicit learning axis involves regions directly adjacent to primary sensory and motor regions, consistent with the need for near instantaneous access to information encoded within sensorimotor regions, and the ability of implicit adaptation to occur automatically and outside conscious awareness. By contrast, the explicit learning axis involves regions positioned at the very apex of the hierarchy, in the DMN. This is consistent with explicit learning requiring a high-level understanding of the task structure (i.e., that the cursor has in fact been rotated rather than left/right displaced), which itself requires the integration of information over the timescale of multiple trials (see Fig 7). This hierarchical view accords with functional differences in the time horizons over which sensorimotor versus transmodal areas process information [76]. Previous research demonstrates that temporal windows of information processing change hierarchically across the cortex [77–79], with sensorimotor areas integrating information over milliseconds to seconds, and transmodal areas over seconds to minutes or longer. These longer "temporal receptive windows" in transmodal cortex [77,78] are thought to enable the encoding of more slowly changing states and the incorporation of acquired knowledge into complex scenarios, both of which are crucial for developing cognitive strategies during learning.

As illustrated in Fig 7, various higher-order control signals may alter the functioning of the motor system at different levels of abstraction. Within the timescale of a single trial, control structures can select a target for action and dictate the control policy used to generate the movement. At a higher level, other structures support the accumulation of information over multiple trials, enabling the construction of a cognitive map of the task space, experimentation with response strategies, and signaling explore/exploit shifts in behavior. These processes are essential for developing and using explicit strategies. However, research on explicit motor learning has largely overlooked the characterization of specific underlying processes that support learning in different task contexts. Although our results share similar limitations, they also highlight important future directions that have been largely ignored in the sensorimotor learning literature. Furthermore, they contextualize several previously reported findings—such as the apparent association between visuomotor learning and episodic memory (e.g., [80,81])—by directly implicating default mode subsystems whose role in memory-guided performance is relatively well characterized [41].

A final unique feature of the current study was in exploring the extent to which the neural axes identified during sensorimotor adaptation generalized across other forms of motor learning. Although few neural studies have examined the use of reinforcement mechanisms during motor learning, there is considerable behavioral evidence indicating that explicit processes are particularly crucial in tasks that deny the kinds of sensory feedback that drive error-based adaptation, such as in our reward-based motor task, or the tasks of [19,49,48,82]. In these types of reward-based motor learning tasks, the agent is provided no feedback indicating precisely how, or to what extent, the executed versus optimal movements differ (unlike during error-based learning), and so they must perform some kind of credit assignment in order to decide which feature of the movement to adjust on subsequent trials [83]. In these contexts, subjects will often develop an explicit understanding of the task structure, with their motor responses demonstrating a clear prioritization of the features relevant to performance [48,49]. Consistent with this, the majority of subjects in our reward-based motor task exhibited well-defined learning periods, in which subjects began to more rapidly converge on the correct solution. These more rapid shifts in behavior have been studied in the context of one-shot learning, where research has implicated regions of the DMN [84] in facilitating rapid learning through an explicit record of past action-outcome associations [85]. As noted above, these regions featured prominently in our Explicit neural axis, the expression of which predicted subject performance in the reward-based motor task.

Although our study identifies distinct patterns of connectivity with motor cortex during sensorimotor adaptation and the component processes that these patterns support, it also leaves several important questions for future research. First, because we restricted our analysis to bipartite functional interactions between the motor cortex and the rest of the brain, we cannot speak to connectivity changes occurring outside of the motor cortex and how they may relate to these different learning processes. For example, our analysis does not preclude communication between regions of visual cortex and frontoparietal regions in the monitoring of visuomotor performance. Nevertheless, we believe our study provides a critical first step in understanding how interactions with the motor system drive sensorimotor adaptation. Second, although we included regions of both the striatum and cerebellum in our analyses, we found that they tended to load relatively weakly on the neural axes as compared to cortical regions. This is likely to reflect relatively stronger cortico-cortical functional connectivity (compared to cortico-extracortical connectivity), rather than the non-involvement of these regions in learning. Indeed, as cortico-cerebellar/striatal functional connectivity is relatively weaker, it may simply be less predictively useful in our regression analyses than using cortico-cortical connectivity. Indeed, when we directly examined functional connectivity between the

cerebellum and motor cortex during learning, we found that task-related modulation of connectivity was greatest in motor regions of the cerebellum, consistent with the known role of the cerebellum in successful visuomotor adaptation [86].

## Methods

### Participants

Forty-six right-handed subjects (27 females, aged 18 to 28 years, mean age: 20.3 years) participated in 3 separate testing sessions, each spaced approximately 1 week apart: the first, an MRI-training session, was followed by 2 subsequent MRI experimental sessions. Of these 46 subjects, 10 participants were excluded from the final analysis [1 subject was excluded due of excessive head motion in the MRI scanner (motion greater than 2 mm or 2° rotation within a single scan); 1 subject was excluded due interruption of the scan during the learning phase of the reward-based motor task; 5 subjects were excluded due to poor behavioral performance in one or both tasks (4 of these participants were excluded because >25% of trials were not completed within the maximum trial duration; and one because >20% of trials had missing data due to insufficient pressure of the fingertip on the MRI-compatible tablet); 3 subjects were excluded due to a failure to properly perform the reward-based task (these subjects did not trace the visible rightward path during the baseline phase, but rather continued to trace a straight line for the entire duration of the task, suggesting that they did not attend to the task)]. Right-handedness was assessed with the Edinburgh handedness questionnaire [87]. Participants' informed consent was obtained before beginning the experimental protocol. The Queen's University Research Ethics Board approved the study and it was conducted in coherence to the principles outlines in the Canadian Tri-Council Policy Statement on Ethical Conduct for Research Involving Humans and the principles of the Deceleration of Helsinki (1964).

### Procedure

In the first session, participants took part in an MRI-training session inside a mock (0 T) scanner that was made to look and sound like a real MRI scanner. We undertook this training session to (1) introduce participants to features of the VMR and reward-based motor tasks that would be subsequently performed in the MRI scanner; (2) ensure that subjects obtained baseline performance levels on those 2 tasks; and (3) ensure that participants could remain still for 1.5 h. To equate baseline performance levels across participants, subjects performed 80 trials per task. To train participants to remain still in the scanner, we monitored subjects' head movement via a Polhemus sensor attached to each subject's forehead (Polhemus, Colchester, Vermont). This allowed a real-time read-out of subject head displacement in the 3 axes of translation and rotation (6 dimensions total). Whenever subjects' head translation and/or rotation reached 0.5 mm or 0.5° rotation, subjects received an unpleasant auditory tone, delivered through a speaker system located near the head. All of the subjects used in the study learned to constrain their head movement through this training regimen. Following this first session, subjects then subsequently participated in reward-based VMR motor tasks, respectively (see below for details).

### Apparatus

During testing in the mock (0 T) scanner, subjects performed hand movements that were directed towards a target by applying fingertip pressure on a digitizing touchscreen tablet (Wacom Intuos Pro M tablet). During the actual MRI testing sessions, subjects used an MRI-compatible digitizing tablet (Hybridmojo LLC, California, United States of America). In both

the mock and real MRI scanner, the target and cursor stimuli were rear-projected with an LCD projector (NEC LT265 DLP projector, 1,024 × 768 resolution, 60 Hz refresh rate) onto a screen mounted behind the participant. The stimuli on the screen were viewed through a mirror fixated on the MRI coil directly above the participants' eyes, thus preventing the participant from being able to see their hand.

## Sensorimotor adaptation task

To study sensorimotor adaptation, we used the well-characterized VMR paradigm ([21]; Fig 1). During the VMR task, participants performed baseline trials in which they used their right index finger to perform center-out target-directed movements. After these baseline trials, we applied a 45$^\circ$ clockwise (CW) rotation to the viewed cursor, allowing investigation of VMR learning. Following this, we assessed participants' re-aiming strategy associated with their learning.

Each trial started with the participant moving the cursor (3 mm radius cyan circle) into the start position (4 mm radius white circle) in the center of the screen by sliding the index finger on the tablet. To guide the cursor to the start position, a ring centered around the start position indicated the distance between the cursor and the start position. The cursor became visible when its center was within 8 mm of the center of the start position. After the cursor was held within the start position for 0.5 s, a target (5 mm radius red circle) was shown on top of a gray ring with a radius of 60 mm (i.e., the target distance) centered around the start position. The target was presented at one of 8 locations, separated by 45$^\circ$ (0, 45, 90, 135, 180, 225, 270, and 315$^\circ$), in pseudorandomized bins of 8 trials. Participants were instructed to hit the target with the cursor by making a fast finger movement on the tablet. They were instructed to "slice" the cursor through the target to minimize online corrections during the reach. If the movement was initiated (i.e., the cursor had moved fully out of the start circle) before the target appeared, the trial was aborted and a feedback text message "Too early" appeared centrally on the screen. In trials with correct timing, the cursor was visible during the movement to the ring and then became stationary for 1 s when it reached the ring, providing the participant with visual feedback of their endpoint reach error. If any part of the stationary cursor overlapped with any part of the target, the target colored green to indicate a hit. Each trial was terminated after 4.5 s, independent of whether the cursor had reached the target. After a delay of 1.5 s, allowing time to save the data, the next trial started with the presentation of the start position.

During the participant training session in the mock MRI scanner (i.e., 2-weeks prior to the VMR MRI testing session), participants performed a practice block of 40 trials with veridical feedback (i.e., no rotation was applied to the cursor). This training session exposed participants to several key features of the task (e.g., use of the touchscreen tablet, trial timing, presence, and removal of cursor feedback) and allowed us to establish adequate performance levels.

At the beginning of the MRI testing session, but prior to the first scan being collected, participants re-acquainted themselves with the VMR task by performing 80 practice trials with veridical cursor feedback. Next, we collected an anatomical scan, followed by the baseline fMRI experimental run, wherein subjects performed 64 trials of the VMR task with veridical cursor feedback using their right hand. Next, they performed the learning scan, wherein subjects performed 160 trials in which feedback of the cursor during the reach was rotated clockwise by 45$^\circ$. Following this fMRI experimental run, participants were asked to report their strategic aiming direction (over 16 trials). In these trials, a line between the start and target position appeared on the screen at the start of each trial. Participants were asked to use a separate MRI joystick (Current Designs, Inc.) positioned at their left hip to rotate the line to the

direction that they would aim their finger movement in for the cursor to hit the target, and click the button on the joystick box when satisfied. Following the button click, the trial proceeded as a normal reach trial. Specifically, the following text was displayed on the screen to participants at the onset of the reporting trials: "Use the joystick to rotate the line to the direction that you will aim your finger in for the cursor to hit the target. If you think your finger should aim straight at the target, then leave the line at the target and click the button on the joystick. If you think your finger should aim somewhere else for the cursor to hit the target, rotate the line to your aiming direction, and then click the button. Next, hit the target with your right finger."

These 16 "report" trials were followed by 16 normal rotation trials. As several subjects expressed confusion about the nature of the report trials, often failing to adjust the line at all during the first few trials, we discarded the first 8 report trials for all subjects, and calculated the mean aim direction using the final 8 trials (note, however, that we observed very similar results whether estimates of the explicit and implicit components were based on these 8 versus all 16 report trials, see S2 Fig). Note that we had subjects perform these report trials following learning (as in [49]) given our prior behavioral work showing that the inter-mixing of report trials during learning can lead to more participants adopting an explicit, re-aiming strategy [22,49], thereby distorting participants' learning curves. Participants were not informed about nature or presence of the visuomotor rotation before or during the experiment, and received no "coaching" about how to specifically perform the reporting trials (we adopted this "hands-off" approach given our concern that any experimenter instructions or invention might bias subjects' learning and reporting behavior).

Note that all subjects performed the VMR task after having performed the reward-based motor task (i.e., testing order was not counterbalanced across the RL and EL tasks). We made this decision in light of our knowledge that subjects may alter their behavior after performing explicit report trials in the VMR task [22] (i.e., often developing explicit awareness of the perturbation and using that to guide their performance). We were concerned that, had subjects performed the VMR task first, they may have anticipated some experimental manipulation (or deception), and thus would not have approached the reward-based motor task in a naive fashion. From this perspective, our approach of having defined the explicit and implicit neural axes on the (second) VMR task and then using those neural axes to predict performance on the (first) reward-based motor task, strengthens our interpretations of the effects.

## Reward-based learning task

In the reward-based motor task (Fig 5, left), participants learned to produce, through reward-based feedback, finger movement trajectories with a specific (unseen) shape. Specifically, subjects were instructed to repeatedly trace, without visual cursor feedback of their actual finger paths, a subtly curved path displayed on the screen (the visible path). During learning trials, participants were told that, following each trial, they would receive a score based on how accurately they traced the visible path (and were instructed to maximize points across trials). However, unbeknownst to them, they actually received points based on how well they traced the mirror-image path (the target path). Critically, because participants received no visual feedback about their actual finger trajectories or the "rewarded" shape, they could not use error-based information to guide learning. Our task was inspired by the motor learning tasks developed by [48,82].

Each trial started with the participant moving the cursor (3 mm radius cyan circle) into the start position (4 mm radius white circle) at the bottom of the screen by sliding their index finger on the tablet. The cursor was only visible when it was within 30 mm of the start position.

After the cursor was held within the start position for 0.5 s, the cursor disappeared and a curved path and a target distance marker appeared on the screen. The target distance marker was a horizontal red line (30 × 1 mm) that appeared 60 mm above the start position. The visible path connected the start position and target distance marker and had the shape of a half sine wave with an amplitude of 0.15 times the target distance. Participants were instructed to trace the curved visible path. When the cursor reached the target marker distance, the marker changed color from red to green to indicate that the trial was completed. Importantly, participants did not receive feedback about the position of their cursor during the trial.

In the baseline block, participants did not receive feedback about their performance. In the learning block, participants were rewarded 0 to 100 points after reaching the target distance marker, and participants were instructed to do their best to maximize this score across trials (the points were displayed as text centrally on the screen). They were told that to increase the score, they had to "trace the line more accurately." Each trial was terminated after 4.5 s, independent of whether the cursor had reached the target. After a delay of 1.5 s, allowing time to save the data, the next trial started with the presentation of the start position.

To calculate the score on each trial in the learning block, the x position of the cursor was interpolated at each cm displacement from the start position in the y direction (i.e., at exactly 10, 20, 30, 40, 50, and 60 mm). For each of the 6 y positions, the absolute distance between the interpolated x position of the cursor and the x position of the rewarded path (mirror image of visible path) was calculated. The sum of these errors was scaled by dividing it by the sum of errors obtained for a half cycle sine-shaped path with an amplitude of 0.5 times the target distance, and then multiplied by 100 to obtain a score ranging between 0 and 100. The scaling worked out so that a perfectly traced visible path would result in an imperfect score of 40 points. This scaling was chosen on the basis of extensive pilot testing in order to achieve variation in subject performance, and to ensure that subjects still received informative score feedback when tracing in the vicinity of the visible trajectory.

During the participant training session in the mock MRI scanner (i.e., 1-week prior to the MRI testing session), participants only performed a practice block in which they traced a straight line with (40 trials) and then without (40 trials) visual feedback of the position of the cursor during the reach. As with the VMR task, this training session exposed participants to several key features of the task (e.g., use of the touchscreen tablet, trial timing, used of cursor feedback to correct for errors) and allowed us to establish adequate baseline performance levels. Importantly, however, this training session did not allow for any reward-based learning to take place.

At the beginning of the MRI testing session, but prior to the first scan being collected, participants re-acquainted themselves with the reward-based motor task by first performing a practice block in which they traced a straight line with (40 trials) and then without (40 trials) visual feedback of the position of the cursor during the reach. Next, we collected an anatomical scan, following by a DTI scan, followed by a resting-state fMRI scan. During the latter resting-state scan, participants were instructed to rest with their eyes open while fixating on a central cross location presented on the screen. Following this, participants performed the reward-based motor task, which consisted of 2 separate experimental runs without visual feedback of the cursor: (1) a baseline block of 70 trials in which they attempted to trace the curved visible path and no score was provided; and (2) a separate learning block of 200 trials in which participants were instructed to maximize their score shown at the end of each trial.

## Behavioral data analysis

**VMR task analysis.**    Trials in which the reach was initiated before the target appeared (4% of trials) or in which the cursor did not reach the target within the time limit (5% of trials)

were excluded from the offline analysis of hand movements. As insufficient pressure on the touchpad resulted in a default state in which the cursor was reported as lying in the top left corner of the screen, we also excluded trials in which the cursor jumped to this position before reaching the target region (2% of trials). We then applied a conservative threshold on the movement and reaction times, removing the top 0.05% of trials across all subjects. As the VMR task required the subject to determine the target location prior to responding, we also set a lower threshold of 100 ms on the reaction time.

**Reward-based motor task analysis.** Trials in which the cursor did not reach the target distance marker within the time limit were excluded from the offline analysis of hand movements (1% of trials). As in the VMR task, we excluded trials in which insufficient pressure was applied (2% of trials), and also separately applied a conservative threshold on the movement and reaction times, removing the top 0.05% of trials across all subjects. As the VMR task did not involve response discrimination, we did not set a lower threshold on these variables.

For each trial, we computed a movement score by integrating the horizontal position of the trajectory, so as to derive a scalar measure of the overall leftward or rightwardness of the movement. Note that we flipped the sign of this measure so that positive (resp. negative) values denote trajectories closer to the target (resp. trained) trajectory. To each subjects' movement data, we fit a generalized sigmoid function in which the predicted movement score on each trial $x$ was given by

$$f(x) = L + \frac{U - L}{1 + \exp[-k(x - m)]} \tag{1}$$

where $L$ and $U$ denote the lower and upper asymptotes, respectively; $m$ denotes the midpoint of the learning period; and $k$ denotes the learning rate (but see our comments below). Because the fits were unstable for several subjects with very slow learning rates, we regularized the fits with the following Bayesian model:

$$y_i \sim \text{Normal}(\mu_i \sigma^2)$$

$$\mu_i = L + \frac{U - L}{1 + \exp[-k(x - m)]}$$

$$m \sim \text{Normal}(1.7, 0.5)$$

$$L \sim \text{Normal}(-0.2, 0.25)$$

$$U \sim \text{Normal}(0, 0.25)$$

$$k \sim \text{Normal}(7, 2)$$

$$\sigma \sim \text{Normal}^+(0, 0.25);$$

Note that trial numbers were scaled by 100 to keep the units comparable, so that the prior mean of 1.7 on the midpoint $m$ was centered at trial 170 (i.e., the center of the learning block). The prior mean of $-0.25$ on $L$ was centered about the movement score corresponding to tracing the initial learned trajectory, while the prior mean on $U$ corresponds to the midpoint between perfect learning and an absence of learning. The prior scale of 0.25 on the error standard deviation was chosen empirically to place high probability mass over the standard deviations observed in subjects' baseline blocks.

Models were fit using the Hamiltonian Monte-Carlo algorithm implemented by the Stan programming language [88]. For each subject, we fit the model using 3 chains of 5,000 samples, of which the first 1,500 were discarded as burn-in. In each case, visual inspection of the chains suggested adequate convergence, with potential scale reduction factor $\hat{R}$ less than 1.01 for all parameters and all subjects.

For each subject, pre-learning and learning epochs were defined using the posterior mean parameter estimates. Specifically, we defined the end of the pre-learning epoch to be the point at which the posterior mean curve exceeded 2.5% of its total range $U - L$. The beginning of the learning epoch was then defined to be the point at which the curve exceeded 5% of its total range. Using these boundaries, we then defined pre-learning and learning periods of 100 imaging volumes by setting the end of the pre-learning period to be the 0.025 quantile of the sigmoid fit, and the beginning of the post-learning period to be the 0.975 quantile. See S3 Fig for fits to individual subjects.

Visual inspection of the model fits indicated that the sigmoid generally accurately captured the overall pattern of subjects' learning, with a few exceptions. In S3 Fig, we note 3 subjects who did not display well-defined learning periods, and for whom the fitted sigmoid was nearly flat. In these cases, we retained the identified pre-learning and learning epochs, and these subjects were simply recorded as having extremely low learning rates. Two subjects were poorly characterized by the sigmoid fit, and so we manually identified pre-learning and learning epochs in these cases. Each of these subjects is indicated in the supplemental figure. In these latter 2 subjects, the parameter k did not encode the learning rate within the learning epoch itself, and so for each subject we defined their learning rate (for the analysis reported in Fig 5) to be the slope of the line of best fit to the scores in their learning epoch.

## MRI acquisition

Participants were scanned using a 3-Tesla Siemens TIM MAGNETOM Trio MRI scanner located at the Centre for Neuroscience Studies, Queen's University (Kingston, Ontario, Canada). Functional MRI volumes were acquired using a 32-channel head coil and a T2*-weighted single-shot gradient-echo echo-planar imaging (EPI) acquisition sequence (time to repetition (TR) = 2,000 ms, slice thickness = 4 mm, in-plane resolution = 3 mm × 3 mm, time to echo (TE) = 30 ms, field of view = 240 mm × 240 mm, matrix size = 80 × 80, flip angle = 90˚, and acceleration factor (integrated parallel acquisition technologies, iPAT) = 2 with generalized auto-calibrating partially parallel acquisitions (GRAPPA) reconstruction. Each volume comprised 34 contiguous (no gap) oblique slices acquired at a 30˚ caudal tilt with respect to the plane of the anterior and posterior commissure (AC-PC), providing whole-brain coverage of the cerebrum and cerebellum. Each of the task-related scans included an additional 8 imaging volumes at both the beginning and end of the scan. On average, each of the MRI testing sessions lasted 2 h.

At the beginning of the reward-based motor task MRI testing session, a T1-weighted ADNI MPRAGE anatomical was also collected (TR = 1,760 ms, TE = 2.98 ms, field of view = 192 mm × 240 mm × 256 mm, matrix size = 192 × 240 × 256, flip angle = 9˚, 1 mm isotropic voxels). This was followed by a series of Diffusion-Weighted scans, wherein we acquired 2 sets of whole-brain diffusion-weighted volumes (30 directions, b = 1,000 s mm-2, 65 slices, voxel size = 2 × 2 × 2 mm3, TR = 9.3 s, TE = 94 ms) plus 2 volumes without diffusion-weighting (b = 0 s mm-2). Next, we collected a resting-state scan, wherein 300 imaging volumes were acquired. For the baseline and learning scans during the motor task, 222 and 612 imaging volumes were acquired, respectively.

At the beginning of the VMR MRI testing session, we gathered high-resolution whole-brain T1-weighted (T1w) and T2-weighted (T2w) anatomical images (in-plane resolution 0.7 × 0.7

mm2; 320 × 320 matrix; slice thickness: 0.7 mm; 256 AC-PC transverse slices; anterior-to-posterior encoding; 2 × acceleration factor; T1w TR 2,400 ms; TE 2.13 ms; flip angle 8˚; echo spacing 6.5 ms; T2w TR 3,200 ms; TE 567 ms; variable flip angle; echo spacing 3.74 ms). These protocols were selected on the basis of protocol optimizations designed by [89]. Following this, for the baseline and learning functional scans, 204 and 492 imaging volumes were acquired, respectively.

## fMRI preprocessing

Results included in this manuscript come from preprocessing performed using *fMRIPrep* 1.4.1 ([90,91]; RRID:SCR_016216), which is based on *Nipype* 1.2.0 ([92,93]; RRID:SCR_002502).

**Anatomical data preprocessing.** A total of 2 T1-weighted (T1w) images were found within the input BIDS data set. All of them were corrected for intensity non-uniformity (INU) with N4BiasFieldCorrection [94], distributed with ANTs 2.2.0 ([95], RRID:SCR_004757). The T1w-reference was then skull-stripped with a *Nipype* implementation of the antsBrainExtraction.sh workflow (from ANTs), using OASIS30ANTs as target template. Brain tissue segmentation of cerebrospinal fluid (CSF), white-matter (WM), and gray-matter (GM) was performed on the brain-extracted T1w using fast FSL 5.0.9, RRID:SCR_002823, [96]. A T1w-reference map was computed after registration of 2 T1w images (after INU-correction) using mri_robust_template (FreeSurfer 6.0.1, [97]). Brain surfaces were reconstructed using recon-all (FreeSurfer 6.0.1, RRID:SCR_001847, [98]), and the brain mask estimated previously was refined with a custom variation of the method to reconcile ANTs-derived and FreeSurfer-derived segmentations of the cortical gray-matter of Mindboggle (RRID:SCR_002438, [99]). Volume-based spatial normalization to 2 standard spaces (MNI152NLin6Asym, MNI152NLin2009cAsym) was performed through nonlinear registration with antsRegistration (ANTs 2.2.0), using brain-extracted versions of both T1w reference and the T1w template. The following templates were selected for spatial normalization: FSL's MNI ICBM 152 nonlinear 6th Generation Asymmetric Average Brain Stereotaxic Registration Model ([100], RRID: SCR_002823; TemplateFlow ID: MNI152NLin6Asym), ICBM 152 Nonlinear Asymmetrical template version 2009c ([101], RRID:SCR_008796; TemplateFlow ID: MNI152NLin2009cAsym).

**Functional data preprocessing.** For each of the BOLD runs per subject (across all tasks and sessions), the following preprocessing was performed. First, a reference volume and its skull-stripped version were generated using a custom methodology of *fMRIPrep*. The BOLD reference was then co-registered to the T1w reference using bbregister (FreeSurfer) which implements boundary-based registration [102]. Co-registration was configured with 9 degrees of freedom to account for distortions remaining in the BOLD reference. Head-motion parameters with respect to the BOLD reference (transformation matrices and 6 corresponding rotation and translation parameters) are estimated before any spatiotemporal filtering using mcflirt (FSL 5.0.9, [103]). BOLD runs were slice-time corrected using 3dTshift from AFNI 20160207 ([104], RRID:SCR_005927). The BOLD time-series were resampled to surfaces on the following spaces: *fsaverage*. The BOLD time-series (including slice-timing correction when applied) were resampled onto their original, native space by applying a single, composite transform to correct for head-motion and susceptibility distortions. These resampled BOLD time-series will be referred to as preprocessed BOLD in original space, or just preprocessed BOLD. The BOLD time-series were resampled into several standard spaces, correspondingly generating the following spatially normalized, preprocessed BOLD runs: MNI152NLin6Asym, MNI152NLin2009cAsym. First, a reference volume and its skull-stripped version were generated using a custom methodology of *fMRIPrep*. Automatic removal of motion artifacts using

independent component analysis (ICA-AROMA, [105]) was performed on the preprocessed BOLD on MNI space time-series after removal of non-steady state volumes and spatial smoothing with an isotropic, Gaussian kernel of 6 mm FWHM (full-width half-maximum). Corresponding "non-aggresively" denoised runs were produced after such smoothing. Additionally, the "aggressive" noise-regressors were collected and placed in the corresponding confounds file. Several confounding time-series were calculated based on the preprocessed BOLD: framewise displacement (FD), DVARS and 3 region-wise global signals. FD and DVARS are calculated for each functional run, both using their implementations in *Nipype* (following the definitions by [106]). The 3 global signals are extracted within the CSF, the WM, and the whole-brain masks. Additionally, a set of physiological regressors were extracted to allow for component-based noise correction (*CompCor*, [107]). Principal components are estimated after high-pass filtering the preprocessed BOLD time-series (using a discrete cosine filter with 128s cut-off) for the 2 *CompCor* variants: temporal (tCompCor) and anatomical (aCompCor). tCompCor components are then calculated from the top 5% variable voxels within a mask covering the subcortical regions. This subcortical mask is obtained by heavily eroding the brain mask, which ensures it does not include cortical GM regions. For aCompCor, components are calculated within the intersection of the aforementioned mask and the union of CSF and WM masks calculated in T1w space, after their projection to the native space of each functional run (using the inverse BOLD-to-T1w transformation). Components are also calculated separately within the WM and CSF masks. For each CompCor decomposition, the *k* components with the largest singular values are retained, such that the retained components' time series are sufficient to explain 50 percent of variance across the nuisance mask (CSF, WM, combined, or temporal). The remaining components are dropped from consideration. The head-motion estimates calculated in the correction step were also placed within the corresponding confounds file. The confound time series derived from head motion estimates and global signals were expanded with the inclusion of temporal derivatives and quadratic terms for each [108]. Frames that exceeded a threshold of 0.5 mm FD or 1.5 standardized DVARS were annotated as motion outliers. All resamplings can be performed with a single interpolation step by composing all the pertinent transformations (i.e., head-motion transform matrices, susceptibility distortion correction when available, and co-registrations to anatomical and output spaces). Gridded (volumetric) resamplings were performed using antsApplyTransforms (ANTs), configured with Lanczos interpolation to minimize the smoothing effects of other kernels [109]. Non-gridded (surface) resamplings were performed using mri_vol2surf (FreeSurfer).

Many internal operations of *fMRIPrep* use *Nilearn* 0.5.2 ([110], RRID:SCR_001362), mostly within the functional processing workflow. For more details of the pipeline, see the section corresponding to workflows in *fMRIPrep*'s documentation.

**Region of interest (ROI) extraction.**   Regions of interest were identified using separate parcellations for the cerebellum, basal ganglia, and cortex. For the cerebellum, we extracted all regions using the atlas of Diedrichsen [32]. For the basal ganglia, we extracted the caudate, putamen, accumbens, and pallidum, using the Harvard-Oxford atlas [111,112]. Finally, for cortex, we extracted regions using the 400 region Schaefer parcellation [29]. For the Somatomotor A and B networks, we extracted only those regions which [29] identified as being activated during finger and tongue movement. For all regions and networks, we created time-series data by computing the mean BOLD signal within each ROI.

**Covariance estimation and centering.**   For each subject, we used the Ledoit and Wolf shrinkage estimator [113] to derive covariance matrices for periods of equivalent length (100 imaging volumes) during baseline, and the beginning and end of the learning blocks of the VMR. For the baseline blocks, we extracted 100 imaging volumes equally spaced throughout the scan, so as the capture the entirety of the baseline epoch of the task. For the reward-based

motor task, we derived covariance matrices for the pre-learning, learning, and post-learning epochs based on sigmoid fitting procedure described above.

To center the covariance matrices, we took the approach advocated by [114], which leverages the natural geometry of the space of covariance matrices. We have implemented many of the computations required to replicate the analysis in a publicly available R package **spdm**, which is freely available from a Git repository at https://github.com/areshenk-rpackages/spdm.

The procedure is as follows. Letting $S_{ij}$ denote the $j$'th covariance matrix for subject $i$, we computed the grand mean covariance $\bar{S}$ over all subjects using the fixed-point algorithm described by [115], as well as subject means $\bar{S}_i$. We then projected the each covariance matrix $S_{ij}$ onto the tangent space at $\bar{S}_i$ to obtain a tangent vector

$$T_{ij} = \bar{S}_i^{1/2} log(\bar{S}_i^{-1/2} S_{ij} \bar{S}_i^{-1/2}) \bar{S}_i^{1/2} \tag{2}$$

where log denotes the matrix logarithm. The tangent vector $T_{ij}$ then encodes the difference in covariance between the covariance $S_{ij}$ and the subject mean $\bar{S}_i$. We then transported each tangent vector to the grand mean $\bar{S}$ using the transport proposed by [114], obtaining a centered tangent vector $T_{ij}^c$ given by

$$T_{ij}^c = G T_{ij} G^T \tag{3}$$

where $G = \bar{S}^{1/2} \bar{S}_i^{-1/2}$. This centered tangent vector (a symmetric matrix) encodes the same difference in covariance, but now expressed relative to the mean resting state scan.

**Construction of the VMR learning-related neural axes and prediction of reward-based motor performance.**   Here, we describe our approach for the main analysis—i.e., construction of the (performance corrected) explicit neural axis derived from the contralateral hand area, during the early learning epoch.

For each tangent vector in the early learning epoch, we computed the mean connectivity between the contralateral hand area (within the somatomotor network) and the remaining ROIs. The resulting vectors (containing an entry for each non-motor ROI) were used as the features for our predictive model.

Call the resulting matrix of predictors $\mathbf{X} = [\mathbf{x_1}, \mathbf{x_2}, ..., \mathbf{x_n}]$, where $\mathbf{x_i}$ is a vector containing the mean connectivity of the $i$'th ROI with the hand area of motor cortex. We then computed the explicit learning score for each subject, defined as the circular mean difference between the target location and the subject's reported aim direction. These scores formed the response variable $\mathbf{y}$. Our corrected scores were then derived by orthogonalizing with respect to subjects' VMR performance scores. That is, we regressed $\mathbf{y}$ onto the vector of VMR mean early error and used the residuals for our analysis, and the same was done for each predictor (that is, each column in $\mathbf{X}$). Both the predictors and outcome were then standardized prior to analysis.

We then fit a ridge regression model with separate network-level penalty terms using the model described by [34], and implemented in the R package GRridge [116]. That is, each of the 17 networks comprising the predictors—15 non-motor cortical networks, plus cerebellum and basal ganglia—possessed their own shrinkage parameters. These parameters were separately tuned using an internal leave-one-out cross-validation loop, so that networks whose regions contributed relatively little to overall predictive performance were shrunken more

harshly relative to predictively useful networks. The precise arguments were as follows:

$$\text{grridge}(X, y, \text{partitions} = \text{network\_labels},$$

$$\text{unpenal} =\sim 1, \text{offset} = \text{NULL},$$

$$\text{method} = ''\text{stable}'', \text{niter} = 10,$$

$$\text{innfold} = \text{NULL}, \text{fixedfoldsinn} =$$

$$\text{TRUE},$$

$$\text{selectionEN} = F, \text{cvlmarg} = 1,$$

$$\text{savepredobj} = ''\text{all}'').$$

The resulting parameter vector $\boldsymbol{\beta}$ contains, for each ROI, its estimated conditional contribution to subjects' explicit report.

We then derived, for each subject, the mean functional connectivity of each region with the motor cortex during the learning period of the reward-based motor task (derived from the centered tangent vectors, as described above). The Pearson correlation between this vector and the vector $\boldsymbol{\beta}$ was the subject's Explicit neural axis score.

Note that all other models (such as the derivation of the implicit or performance neural axis), or neural axes using separate areas of motor cortex (e.g., tongue areas) or different time windows (e.g., baseline or pre-learning) were constructed identically, mutatis mutandis (e.g., using implicit scores rather than explicit reports).

## Behavioral control study

Fourteen right-handed paid volunteers (8 females; mean age: 22.3 years) participated in a behavioral control study conducted outside the MRI scanner. This study adhered to the same research ethics approvals and consent procedures as the MRI study. In this behavioral testing, participants performed a VMR task similar in structure and number of trials to the MRI testing session, except for changes in the experimental setup and the inclusion of intermittent reporting trials during the learning phase.

**Apparatus.** During this testing session, participants were seated at a table with their chin and forehead supported by a headrest approximately 50 cm from a vertical LCD monitor ($47.5 \times 26.5$ cm; resolution $1,920 \times 1,080$ pixels) displaying the stimuli. They performed reaching movements by sliding a stylus across a digital drawing tablet (active area $311 \times 216$ mm; Wacom Intuous) placed on the table. The ratio between the stylus tip movement and the cursor movement on the screen was set to 1:2, meaning a 5-cm movement on the tablet resulted in a 10-cm cursor movement. Movement trajectories were sampled at 100 Hz by the digitizing tablet. To prevent visual feedback from the hand and tablet, a piece of black cardboard was attached to the headrest, occluding the view.

**Procedure.** The procedures for this testing session were based on our prior behavioral work in this area [22,49]. Each trial began with the participant moving the stylus with their right hand to a central start position (5 mm radius circle). When the unseen cursor was within 5 cm of the start position, a ring appeared around the start position to indicate the cursor's distance, guiding the participant to move to the start position by reducing the ring's size. The cursor (4 mm radius circle) appeared on the screen when it reached the start position (9 mm distance). After holding the cursor within the start position for 500 ms, a target (6 mm radius open circle) appeared on a 10-cm radius ring around the start position at one of 8 locations, spaced 45˚ apart (i.e., 0, 45, 90, 135, 180, 225, 270, and 315˚). The ring was marked by 64 non-target "landmarks" (3 mm radius outlined circles), spaced 5.625˚ apart. After a 2-s delay, the

target filled in (turned red), cueing the participant to perform a fast "slicing" movement to the target. If the participant started the movement before the cue or more than 1 s after the cue, the trial was aborted with feedback messages indicating "Too early" or "Too late." In correctly timed trials, the cursor was visible during the movement to the ring and became stationary for 1 s upon reaching the ring, providing visual feedback of the endpoint error. If any part of the cursor overlapped with the target, the target turned green to indicate a hit. If the movement took longer than 300 ms, a "Too slow" feedback message appeared on the screen.

To directly assess the contribution of a strategic re-aiming process to learning, participants performed intermittent 'reporting' trials during the learning period. These reporting trials were intermixed within each bin of 8 trials in a 1:4 ratio, meaning 2 of the 8 trials per bin (randomly selected across locations) were reporting trials. In these trials, participants were instructed to report the aiming direction of their right hand before each reach movement by turning a knob with their left hand to rotate a line on the screen, aligning it with their strategic re-aiming direction. Once satisfied with the line's direction (typically taking 2 s on average), the participant clicked a button next to the knob, and the line disappeared. After a 1-s delay, the target filled in, cueing the participant to execute the reach.

**Data analysis.**   Trials where the movement was initiated too early or too late (detected online; 4% of trials) or where the movement duration exceeded 300 ms (4% of trials) were excluded from the analysis. We measured performance (error) by computing the angular difference between the target location and the final cursor position. We then averaged subjects' errors using the 6 non-report trials within each 8 trial block (S2A Fig). We estimated early and late explicit knowledge by computing the mean report angle (relative to the target) during the first and last block of 32 trials, consistent with our definition of the early learning period in our main analysis, and with our report block conducted at the end of learning.

## Supporting information

**S1 Fig. Subjects' explicit re-aiming strategy during early and late learning.**
(TIF)

**S2 Fig. Relationship between explicit and implicit components estimated using different numbers of reporting trials.**
(TIF)

**S3 Fig. Functional connectivity of the cerebellum during VMR task performance.**
(TIF)

**S4 Fig. Individual subject learning curves during the reward-based motor task.**
(TIF)

**S5 Fig. Robustness analysis of different window sizes.**
(TIF)

**S6 Fig. Explicit axis derived from late VMR learning.**
(TIF)

## Acknowledgments

The authors would like to thank Martin York and Sean Hickman for technical assistance, and thank Don O'Brien for assistance with data collection.

## Author Contributions

**Conceptualization:** Corson N. Areshenkoff, Anouk J. de Brouwer, J. Randall Flanagan, Jason P. Gallivan.

**Data curation:** Corson N. Areshenkoff, Anouk J. de Brouwer.

**Formal analysis:** Corson N. Areshenkoff, Anouk J. de Brouwer.

**Funding acquisition:** Jason P. Gallivan.

**Investigation:** Corson N. Areshenkoff, Anouk J. de Brouwer, Joseph Y. Nashed, Jason P. Gallivan.

**Methodology:** Corson N. Areshenkoff, Anouk J. de Brouwer, Joseph Y. Nashed, Jason P. Gallivan.

**Project administration:** Jason P. Gallivan.

**Resources:** Corson N. Areshenkoff, Anouk J. de Brouwer, Daniel J. Gale, J. Randall Flanagan, Jason P. Gallivan.

**Software:** Corson N. Areshenkoff, Daniel J. Gale.

**Supervision:** Jason P. Gallivan.

**Validation:** Corson N. Areshenkoff.

**Visualization:** Corson N. Areshenkoff, Jonathan Smallwood.

**Writing – original draft:** Corson N. Areshenkoff, Jonathan Smallwood, Jason P. Gallivan.

**Writing – review & editing:** Corson N. Areshenkoff, Jonathan Smallwood, J. Randall Flanagan, Jason P. Gallivan.

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
