## [Decision Letter · Decision Letter 0]

27 Jun 2024

Dear Jason, 

Thank you for submitting your manuscript entitled "Distinct patterns of connectivity with motor cortex reflect component processes of sensorimotor learning" for consideration as a Research Article by PLOS Biology.

As you may remember, once we decide to send a manuscript out to review, we ask you first to complete the submission by providing the metadata. I had forgotten to do this in this case but sent your manuscript straight to the reviewers instead. They have now submitted their reports. Unfortunately, before I can continue to discuss these reports with the Academic Editor, I need to ask you to now complete your submission by providing the required metadata. To this end, please login to Editorial Manager where you will find the paper in the 'Submissions Needing Revisions' folder on your homepage. Please click 'Revise Submission' from the Action Links and complete all additional questions in the submission questionnaire.

Once your full submission is complete, your paper will undergo a series of checks in preparation for peer review. After your manuscript has passed the checks, I will discuss the reports with the Academic Editor. To provide the metadata for your submission, please Login to Editorial Manager (https://www.editorialmanager.com/pbiology) within two working days, i.e. by Jun 29 2024 11:59PM.

Kind regards,

Christian

Christian Schnell, PhD

Senior Editor

PLOS Biology

cschnell@plos.org

---

## [Editor Report · Decision Letter 1]

31 Jul 2024

Dear Jason,

Apologies for the long silence from my end! I was hoping to be able to send our decision much earlier and was hoping to hear from the Academic Editor any moment but despite multiple chasers, did not get any response. So I needed to find a new Academic Editor which I have now managed and have finally the decision now ready for you.

Thank you for your patience while we considered your revised manuscript "Distinct patterns of connectivity with motor cortex reflect component processes of sensorimotor learning" for publication as a Research Article at PLOS Biology. Your revised study has been evaluated by the PLOS Biology editors, the Academic Editor and the original reviewers.

In light of the reviews, which you will find at the end of this email, we would like to invite you to revise the work to thoroughly address Reviewer 3's report.

As you will see below, Reviewer 1 and Reviewer 2 were satisfied with your revision. Reviewer 3, in contrast, was not convinced that their concerns have been fully addressed. After discussing this with the Academic Editor, we would encourage you to address Reviewer 3's concerns by:

1. Relating changes in reaction time (a proxy for the discovery of an explicit strategy) to changes in neural activity during both early (strategy discovery) and late learning (strategy implementation) phases.

2. Recognizing that this analysis might be underpowered, perform a subgroup analysis of participants who exhibit different explicit re-aiming behaviors. Consider an unsupervised clustering of explicit re-aiming behaviour (e.g., Figure 5 of Tsay et al; in press at eLife).

3. Justifying your choice and perform sensitivity analyses with varying numbers of volumes and learning trials.

Given the extent of revision needed, we cannot make a decision about publication until we have seen the revised manuscript and your response to the reviewers' comments. Your revised manuscript is likely to be sent for further evaluation by all or a subset of the reviewers.

**IMPORTANT - SUBMITTING YOUR REVISION**

*Re-submission Checklist*

*Published Peer Review*

*PLOS Data Policy*

*Blot and Gel Data Policy*

Sincerely,

Christian

Christian Schnell, PhD

Senior Editor

PLOS Biology

cschnell@plos.org

REVIEWS:

Reviewer 1 (Jonathan Tsay):

Thank you for addressing my concerns. I have no additional comments.

Reviewer 2:

The authors have carefully revised the manuscript according to the reviewers comments. I think it is now suitable for publication. It is a solid paper. Good job.

Reviewer 3:

In this revised manuscript, Areshenkoff and colleagues try to address the methodological gaps pointed out in the previous review. While I still think the question is very interesting, and the results tell a nice story, I'm not convinced by some of the answers and the key gaps aren't resolved, hence the concerns regarding the reliability of the results remain.

1. I find the suggestion that explicit reports were "highly stable throughout the VMR task" highly controversial as it contradicts the entire literature on implicit and explicit learning. Looking into the "stable explicit component" in the new behavioral experiment, it seems like it's not exactly the case. Some participants (e.g. S13) show the expected explicit curve, increasing fast and decreasing slowly after. Some showed no explicit aim at all, some very noisy curves, and only a few showed this unexpected stable aim. While all together it leads to a significant correlation, one should be careful not to overinterpret it. Specifically, as we see here different subgroups of explicit learners. Previous studies have addressed such subgroups and related them to the RTs (e.g. Bromberg et al., 2019, eNeuro). This nicely links to the suggestion by Reviewer 1 to look at the increase in RT from baseline to early learning as a proxy measure of explicit strategy in the early learning phase. While I agree with the authors' concern that RTs correlate with other cognitive variables, it is probably the most direct way to address explicit strategy in this dataset. A curve of the change in RT over trials would be very informative and if it supports the authors' suggestion, that the explicit component is stable, they'll have an argument to justify their approach. However, the higher correlation between the reported aim and RTs during late learning (relative to early learning) suggests that the explicit strategy changes (as expected).

2. Implicit learning kicks in slowly, hence, any correction in early trials must be mostly attributed to the explicit strategy, to the extent that the performance during early learning is probably a better proxy of the explicit strategy used there than the explicit aiming reported at the end of the learning. Hence one can argue that the performance captures the early explicit strategy and the "pure" explicit after correcting for performance captures the change in explicit over learning.

3. While I agree with the authors that the network during the formation of the explicit

learning strategy (during early learning) might be more interesting than that of the implementation of that strategy (during late learning), I think the latter is still interesting and I'm convinced that it will provide a far more meaningful and reliable result as it will directly relate to the measured explicit aim. In the revised manuscript the authors performed this analysis (on the last four blocks of 8 trials) but didn't share the results as they did for early learning (in figures 3 and 4) but only the across-task prediction which is difficult to appreciate without knowing if and how different the networks are.

4. While the authors justify their 100-volume choice based on previous work, I find the fact that this number of volumes is 34 trials slightly concerning. Having blocks of 8 trials it would make more sense to look at 32, which corresponds to 4 complete blocks (as they have done for the control analysis during late learning) or 40 trials for 5 blocks.

---

## [Editor Report · Decision Letter 2]

15 Oct 2024

Dear Jason,

Thank you for your patience while we considered your revised manuscript "Distinct patterns of connectivity with motor cortex reflect component processes of sensorimotor learning" for publication as a Research Article at PLOS Biology. This revised version of your manuscript has been evaluated by the PLOS Biology editors and the Academic Editor.

Based on our Academic Editor's assessment of your revision, we are likely to accept this manuscript for publication, provided you satisfactorily address the following data and other policy-related requests:

* We would like to suggest a different title to improve accessibility: "Distinct patterns of connectivity with the motor cortex reflect different components of sensorimotor learning"

* Please add the links to the funding agencies in the Financial Disclosure statement in the manuscript details.

* Please include information about the form of consent (written/oral) given for research involving human participants. All research involving human participants must have been approved by the authors' Institutional Review Board (IRB) or an equivalent committee, and must have been conducted according to the principles expressed in the Declaration of Helsinki. Please mention this in the methods section and also provide approval number for this study from your institutional review board.

* DATA POLICY:

Regardless of the method selected, please ensure that you provide the individual numerical values that underlie the summary data displayed in the following figure panels as they are essential for readers to assess your analysis and to reproduce it: 1C, 3, 4D, 5DE, 6B and SI6AC

* CODE POLICY

We expect to receive your revised manuscript within two weeks. 

*Published Peer Review History*

*Press*

Sincerely,

Christian

Christian Schnell, PhD

Senior Editor

cschnell@plos.org

PLOS Biology

---

## [Editor Report · Decision Letter 3]

8 Nov 2024

Dear Jason,

Thank you for the submission of your revised Research Article "Distinct patterns of connectivity with the motor cortex reflect different components of sensorimotor learning" for publication in PLOS Biology. On behalf of my colleagues and the Academic Editor, Jonathan Tsay, I am pleased to say that we can in principle accept your manuscript for publication, provided you address any remaining formatting and reporting issues. These will be detailed in an email you should receive within 2-3 business days from our colleagues in the journal operations team; no action is required from you until then. Please note that we will not be able to formally accept your manuscript and schedule it for publication until you have completed any requested changes.

PRESS

Sincerely, 

Christian

Christian Schnell, PhD

Senior Editor

PLOS Biology

cschnell@plos.org